# Klf4 glutamylation is required for cell reprogramming and early embryonic development in mice

Buqing Ye[1], Benyu Liu[1], Lu Hao[1,2], Xiaoxiao Zhu[3,4], Liuliu Yang[1,2], Shuo Wang[1], Pengyan Xia [1], Ying Du[1], Shu Meng[3,4], Guanling Huang[1,2], Xiwen Qin[1,2], Yanying Wang[1], Xinlong Yan[1], Chong Li[1], Junfeng Hao[3], Pingping Zhu[1,2], Luyun He[1,2], Yong Tian [4] & Zusen Fan [1,2]

Temporal and spatial-specific regulation of pluripotency networks is largely dependent on the precise modifications of core transcription factors. Misregulation of glutamylation is implicated in severe physiological abnormalities. However, how glutamylation regulates cell reprogramming and pluripotency networks remains elusive. Here we show that cytosolic carboxypeptidases 1 (CCP1) or CCP6 deficiency substantially promotes induced pluripotent cell (iPSC) induction and pluripotency of embryonic stem cells (ESCs). Klf4 polyglutamylation at Glu381 by tubulin tyrosine ligase-like 4 (TTLL4) and TTLL1 during cell reprogramming impedes its lysine 48-linked ubiquitination and sustains Klf4 stability. Klf4-E381A knockin mice display impaired blastocyst development and embryonic lethality. Deletion of TTLL4 or TTLL1 abrogates cell reprogramming and early embryogenesis. Thus, Klf4 polyglutamylation plays a critical role in the regulation of cell reprogramming and pluripotency maintenance.

[1] Key Laboratory of Infection and Immunity of CAS, CAS Center for Excellence in Biomacromolecules, Institute of Biophysics, Chinese Academy of Sciences, Beijing, 100101, China. [2] University of Chinese Academy of Sciences, Beijing, 100049, China. [3] Laboratory Animal Center, Institute of Biophysics, Chinese Academy of Sciences, Beijing, 100101, China. [4] Key Laboratory of RNA Biology, Institute of Biophysics, Chinese Academy of Sciences, Beijing, 100101, China. Buqing Ye, Benyu Liu, Lu Hao and Xiaoxiao Zhu contributed equally to this work.   Correspondence and requests for materials should be addressed to Y.T. (email: ytian@ibp.ac.cn) or to Z.F. (email: fanz@moon.ibp.ac.cn)

Reprogramming resets differentiated somatic cells to a pluripotent state, which can be achieved by nuclear transfer, cell fusion, and transduction of transcription factors[1]. Somatic cells can be reprogrammed to induced pluripotent cells (iPSCs) by expressing pluripotency factors Oct4, Sox2, Klf4, and c-Myc (termed OSKM)[2,3]. The generation of iPSCs can be derived from patient tissues and has great potential for regenerative medicine and cell replacement therapies[4,5]. Several hurdles, including low frequency of iPSC induction and genomic instability of iPSCs, need to be solved prior to development of a safe iPSC technology. However, the molecular mechanisms underlying reprogramming still remain ill-defined. The temporal and spatial-specific regulation of pluripotency networks largely depends on precise modifications and interaction controls of the core transcriptional factors[6–9]. These reprogramming factors are highly modified post-transcriptionally at the levels of mRNA stability, translation and protein activity[7,10].

Protein post-translational modifications (PTMs) such as phosphorylation, acetylation, glycosylation, and ubiquitination play important roles in the regulation of activities of target proteins by changing their chemical or structural properties[11,12]. In-depth quantitative and dynamic proteomic studies reveal that PTMs occur on core transcription factors during the process of pluripotency maintenance and reprogramming[7]. Transcriptional and DNA-binding activities of Oct4 and Sox2 are regulated by phosphorylation, which exert considerable effect on pluripotency maintenance and iPSC generation[7,13]. Acetylation of Sox2 is critical for pluripotency control by regulating its nuclear export and protein stability[14]. O-GlcNacylation directly regulates transcriptional activities of Oct4 and Sox2 in maintaining pluripotency and cell reprogramming[9,15]. Moreover, ubiquitination of Klf4 and Oct4 modulates their half-life and subsequent protein stability[16,17]. It has been reported that B cells treated with C/EBPα can be efficiently reprogrammed into iPSCs by OSKM induction through enhancing chromatin accessibility and Klf4 stability[18]. Therefore, PTMs of reprogramming factors play critical roles in determining the cell fate decision of stem cells.

Glutamylation, a unique PTM, adds glutamate side chains onto the γ-carboxyl groups of glutamic acid residues in the primary sequence of target proteins[19–21]. Polyglutamylation of tubulins, well-known targets of glutamylation, regulates the interaction between microtubules (MTs) and their partners[19], modulating MT-related processes such as ciliary motility, neurite outgrowth and neurodegeneration[21–24]. Glutamylation is catalyzed by polyglutamylases, also called as tubulin tyrosine ligase-like (TTLL) enzymes[25,26]. Glutamylation is a reversible modification that can be hydrolyzed by a family of cytosolic carboxypeptidases (CCPs)[23]. Misregulation of glutamylation causes several physiological abnormalities. CCP1 deficiency causes hyperglutamylation of tubulins, resulting in Purkinje cell degeneration[23,27,28]. We recently demonstrated that CCP6 deficiency induces hyperglutamylation of Mad2, leading to underdevelopment of megakaryocytes and abnormal thrombocytosis[29]. In addition, mutations of TTLL polyglutamylases are also implicated in severe disorders such as retinal dystrophy[30] and pancreatic oncogenesis[31]. However, how glutamylation regulates cell reprogramming and pluripotency maintenance remains elusive.

The Kruppel-like factor (Klf4), together with other three Yamanaka pluripotency factors, is able to reprogram adult fibroblasts into iPSCs[2,3,32], which initiates somatic gene suppression in an early phase and pluripotency gene activation in a late phase during reprogramming[33,34]. Klf4 is highly expressed in mouse embryonic stem cells (ESCs) and rapidly downregulated in the early stage of differentiation[35]. Klf4 can be phosphorylated by ERKs, whose phosphorylation recruits βTrCPs (components of a ubiquitin E3 ligase) to ubiquitinate Klf4 for its degradation[16]. However, the precise modification crosstalk of Klf4 in reprogramming has not clearly defined yet.

In this study, we show that CCP1 or CCP6 deficiency substantially promotes iPSC generation. Klf4 is polyglutamylated by TTLL4 or TTLL1 that impedes its lysine (K) 48-linked ubiquitination to maintain Klf4 stability. Deletion of TTLL4 or TTLL1 impairs cell reprogramming and early embryogenesis.

## Results

**CCP6 or CCP1 deficiency promotes somatic cell reprogramming.** We previously demonstrated that *Ccp6* (official gene name *Agbl4*, referred to here as *Ccp6)*-deficient mice exhibit underdevelopment of megakaryocytes and abnormal thrombocytosis[29]. We next wanted to further explore whether glutamylation modifications were involved in the regulation of cell reprogramming. We noticed that CCP6 deficiency caused higher litter size at birth (Supplementary Figure 1a), whereas Ccp6-deficient mice displayed similar sperm and ovulation numbers compared with littermate control mice, suggesting CCP6 could be implicated in the modulation of cellular reprogramming. To determine whether glutamylation is involved in somatic cell reprogramming, we first generated $Ccp1^{-/-}$ (official gene name *Agtpbp1*, referred to here as *Ccp1*), or $Ccp6^{-/-}$ mouse embryonic fibroblasts (MEFs) (Fig. 1a), and transduced Yamanaka factors OSKM for iPSC induction assays. We noticed that $Ccp1^{-/-}$ and $Ccp6^{-/-}$ MEFs produced more iPSC colonies compared with WT MEFs (Fig. 1b). These iPSCs generated from $Ccp1^{-/-}$ or $Ccp6^{-/-}$ MEFs expressed elevated pluripotent stem cell surface marker SSEA-1 and increased pluripotent genes without partial differentiation (Fig. 1c, and Supplementary Figure 1b, c). These reprogrammed iPSCs expressed similar levels of pluripotent genes compared to ESCs, validating the reprogramming system was efficient (Supplementary Figure 1b). Moreover, the iPSCs derived from $Ccp1^{-/-}$ or $Ccp6^{-/-}$ MEFs were capable of forming teratomas containing cells of all three germ layers and displayed faster growth rates (Fig. 1d). Importantly, *Ccp1* and *Ccp6* double knockout (DKO) MEFs showed higher reprogramming efficiency (Fig. 1b), as well as pluripotent gene expression than $Ccp1^{-/-}$ or $Ccp6^{-/-}$ MEFs alone (Supplementary Figure 1b). In addition, restoration of CCP1 or CCP6 into $Ccp1^{-/-}$ or $Ccp6^{-/-}$ MEFs still went back to a low frequency of iPSC generation (Fig. 1b and Supplementary Figure 1b). Of note, deficiency of CCP1 or CCP6 in feeder-free iPSCs really exhibited elevated pluripotent gene expression, whereas deficiency of CCP1 or CCP6 in MEFs alone did not affect pluripotent gene expression (Supplementary Figure 1d). Overall, these results indicate that deficiency of CCP1 or CCP6 promotes somatic cell reprogramming.

To further determine the physiological role of CCP1 and CCP6 in the process of reprogramming, we silenced CCP1 and CCP6 expression in MEFs with transfection of OSKM, and found CCP1 and CCP6 depletion enhanced alkaline phosphatase (AP)-positive iPSC colony formation and pluripotent gene expression (Supplementary Figure 1e-g). By contrast, overexpression of CCP1 and CCP6 impaired iPSC colony formation as well as downregulated pluripotent gene expression (Supplementary Figure 1e-g). Of note, depletion and overexpression of CCP1 and CCP6 in MEFs did not affect growth rates of MEFs (Supplementary Figure 1h). We also treated MEFs with CCP family protein agonist CoCl$_2$[36] and inhibitor phenanthroline[23] after OSKM induction. Consistently, the agonist CoCl$_2$ abrogated iPSC formation, whereas the inhibitor phenanthroline remarkably enhanced iPSC generation (Fig. 1e and Supplementary Figure 1i). These data further confirm that loss of CCP1 or CCP6 virtually enhances cell reprogramming.

Fertilization initiates cellular reprogramming in zygote and subsequent blastocyst development, which also requires the establishment of pluripotency[37,38]. Since homozygous $Ccp1$-deficient mice are male sterile[27], we then assessed the effect of $Ccp6$ deficiency on blastocyst development. We isolated 2-cell-stage embryos from $Ccp6^{+/+}$ and $Ccp6^{-/-}$ pregnant mice and cultured the embryos ex vivo to allow following development. CCP6 deficiency substantially promoted blastocyst development at embryo day (E) 3.5 compared to WT mice (Fig. 1f). In parallel, expression levels of pluripotent genes were dramatically increased in $Ccp6^{-/-}$ embryos (Supplementary Figure 1j-k). To further determine the physiological relevance of glutamylation, we isolated 2-cell-stage embryos from C57BL6/J mice and cultured them ex vivo with $CoCl_2$ and phenanthroline treatment. We observed that the agonist $CoCl_2$ treatment overtly inhibited embryonic development, while the inhibitor phenanthroline treatment remarkably enhanced blastocyst development (Fig. 1g). Taken together, glutamylation regulations are implicated in the early embryonic development.

**Glutamylation is required for pluripotency maintenance.** We further tested whether CCP1 or CCP6 regulates self-renewal and pluripotency maintenance in ES cells. We silenced CCP1 or CCP6

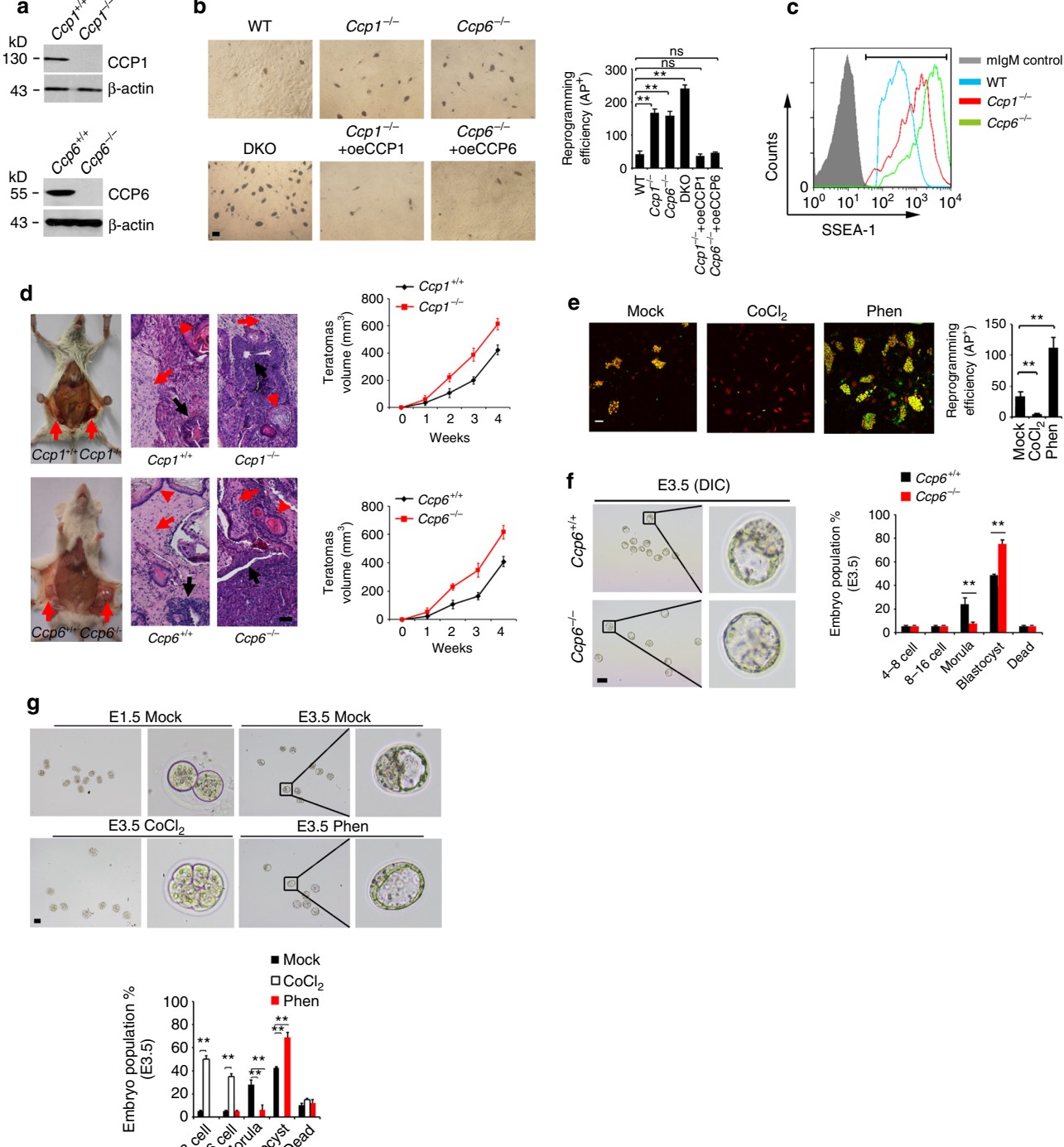

in mouse ESC D3 cells, and noticed that CCP1- or CCP6-depleted D3 cells exhibited increased colony numbers with strong AP-positive staining and upregulated pluripotent gene expression compared with control shRNA (shCtrl)-treated cells (Fig. 2a and Supplementary Figure 2a). Double knockdown of CCP1 with CCP6 induced much more colony numbers than that of CCP1 or CCP6 knockdown alone. By contrast, ectopic expression of CCP1 or CCP6 substantially declined AP-positive colony numbers and downregulated pluripotent gene expression (Fig. 2b and Supplementary Figure 2b). In parallel, CCP1 or CCP6 overexpressing ES R1 cells displayed similar observations (Supplementary Figure 2c). Of note, we noticed that depletion of CCP1 or CCP6 promoted ESC proliferation (Supplementary Figure 2d), whereas overexpression of CCP1 or CCP6 suppressed ESC proliferation (Supplementary Figure 2d). We then measured the glutamylation dynamic changes during the process of early embryonic development. GT335 antibody specifically recognizes the branching point of glutamate side chains and detects all glutamylated forms of target proteins[39]. We observed that GT335 staining signals were elevated during early embryonic development and GT335 staining signals mainly resided in the nucleus of 2-cell and 4–8-cell-stage embryos (Fig. 2c). TTLL members add glutamylation modifications and CCP members remove it of their substrates[19,23]. We observed that polyglutamylases TTLL1 and TTLL4 were highly expressed undergoing early embryonic development (Supplementary Figure 2e). However, both CCP1 and CCP6 were downregulated in 8-cell-stage embryos (Supplementary Figure 2f).

We also performed transcriptome profile assays for CCP6-depleted and shCtrl-treated ESCs. We noticed that CCP6 knockdown in ESCs caused upregulation of pluripotency transcriptional network (Supplementary Figure 2g). In addition, we analyzed RNAseq data set GSE45352[40] for OSKM-induced reprogramming. We found that *Ccp1* was downregulated and *Ttll4* was upregulated over doxycycline-induced OSKM expression (Supplementary Figure 2h). In addition, from RNAseq data set GSE52396[41], *Ccp6* was downregulated during early reprogramming induction (Supplementary Figure 2h). These data suggest that glutamylation is surely involved in the regulation of cell reprogramming.

Intriguingly, similar observations were achieved in CCP1 or CCP6-silenced human ESC H9 cells and human iPSCs derived from OSKM-transduced urothelial cells (Fig. 2d,e and Supplementary Figure 2i-j). Similarly, polyglutamylases TTLL1 and TTLL4 were also highly expressed during human urothelial cell reprogramming (Supplementary Figure 2k). CCP1 and CCP6 were downregulated in human iPSCs (Supplementary Figure 2l). Moreover, CCP1- or CCP6-depleted human iPSCs were able to form teratomas with three germ layer differentiation potential

and grew faster compared with shCtrl-treated iPSCs (Fig. 2f). Whereas CCP1 or CCP6 overexpressing human iPSCs impaired AP-positive colony generation and lost teratoma formation (Fig. 2f). These data suggest that glutamylation also regulates ESC and iPSC pluripotency that is evolutionarily conserved in human reprogramming.

**Klf4 undergoes polyglutamylation in cell reprogramming**. We then measured the glutamylation dynamic changes during the process of somatic cell reprogramming. We observed that GT335 staining signals were elevated over OSKM transduction (Fig. 3a,b). Furthermore, GT335 staining signals mainly resided in the nucleus of MEFs (Fig. 3a). We observed that polyglutamylases TTLL1 and TTLL4 were highly expressed during OSKM-induced reprogramming of MEFs (Supplementary Figure 3a), whereas CCP1 and CCP6 were almost undetectable in mouse iPSCs (Supplementary Figure 3b). These observations were consistent with those of early-stage embryos (Fig. 2c, Supplementary Figure 2e-f). The specificity of GT335 antibody we used was verified (Supplementary Figure 3c). Thus, we proposed that TTLL1 and TTLL4 may mediate substrate glutamylation that is required for iPSC induction.

We next wanted to identify candidate substrates for cell reprogramming. We noticed that *Ccp6*[−/−] MEF lysates with OSKM transduction displayed an around 60 kDa strong differential band compared to those of *Ccp6*[+/+] MEFs (Supplementary Figure 3d). We then generated an enzymatically inactive mutant of CCP6 (CCP6mut) via H230S and E233Q mutations[29], and performed affinity chromatography with CCP6mut-immobilized Affi-gel10 resin. Interestingly, the around 60 kDa band was identified to be Klf4 (Fig. 3c and Supplementary Figure 3e), a novel candidate substrate for CCP6 during cell reprogramming.

Expectedly, Klf4 glutamylation signals were increased after OSKM transduction via immunoprecipitation with immunoblotting assays (Fig. 3d and Supplementary Figure 4a). Since B3 antibody can detect a side chain with two or more glutamate residues[42], we then used this antibody to determine mono-glutamylation or polyglutamylation of glutamylated Klf4. We noticed that B3 antibody could detect glutamylated signals of Klf4 (Supplementary Figure 4b), indicating that glutamylated Klf4 is polyglutamylation. Glutamylation modifications are added at potential acceptor sites with conserved glutamate-rich stretches and flexible environment to be susceptible for catalysis[43]. Klf4 has four putative conserved glutamate-rich regions for glutamylation modifications (Supplementary Figure 4c). We then mutated the first glutamic residue in each putative region to alanine and generated four mutated proteins for in vitro glutamylation assays. We found that only Klf4 Glu381Ala mutant abolished TTLL4-mediated Klf4 polyglutamylation (Fig. 3e), indicating Glu381 is

**Fig. 1** CCP6 or CCP1 deficiency promotes cell reprogramming. **a** *Ccp1*- or *Ccp6*- deficient MEFs were lyzed for immunoblotting. β-actin was used as a loading control. **b** WT or CCPs-deficient MEFs were infected by OSKM factors containing retrovirus and cultured in ESC media for 3 weeks. Alkaline phosphatase (AP)-positive colony numbers per $10^4$ cells were calculated and shown as means ± S.D. **, $P < 0.01$. Scale bar, 50 μm. $n = 5$. **c** SSEA-1 was assayed by flow cytometry in iPSCs derived from *Ccp1*[−/−] or *Ccp6*[−/−] MEFs. **d** iPSCs from *Ccp1*[−/−] or *Ccp6*[−/−] MEFs were injected subcutaneously into NOD/SCID mice. One month later, teratomas were collected for H&E staining. Teratomas volumes were calculated and shown as means ± S.D. (left panel). Red arrow denotes teratomas in left panels. Black arrow, endoderm; red arrow, mesoderm; red arrowhead, ectoderm. Scale bar, 200 μm. Total 6 teratomas from 6 mice were analyzed pre condition ($n = 6$). The teratomas were from 2 iPSC cell lines. **e** 4F2A MEFs were treated with doxycycline (dox) (2 μg/ml), together with CoCl$_2$ (10 μM) or phenanthroline (Phen, 1 μM) in ESC media for iPSC formation as in **b**. Reprogramming efficiency was assayed by Nanog staining after dox removal. Scale bar, 50 μm. Nanog-positive colony numbers per $10^4$ cells were calculated and shown as means ± S.D.**$P < 0.01$. $n = 5$. **f** CCP6 deficiency promoted blastocyst development. Embryos were isolated at E1.5 stage and cultured in KSOM media supplied with 1 mg/ml BSA at 37 °C for 48 h. Distribution of embryonic stage at E3.5 was counted as mean ± S.D. (right panel), **$P < 0.01$. Scale bar, 100 μm. 105 *Ccp6*[+/+] embryos and 122 *Ccp6*[−/−] embryos were observed. $n = 4$. **g** Embryos were isolated at E1.5 stage and cultured in KSOM media with CoCl$_2$ (10 μM) or phenanthroline (1 μM). Distribution of embryonic stage at E3.5 was counted as mean ± S.D (right panel), **$P < 0.01$. Scale bar, 100 μm. 131 mock-treated embryos, 118 CoCl$_2$-treated embryos and 132 phen-treated embryos were observed. $n = 4$. Student's *t* test was used as statistical analysis. oe overexpression, ns no significance

the acceptor site for polyglutamylation. Klf4 Glu381Ala mutant also abolished TTLL1-mediated Klf4 glutamylation (Supplementary Figure 4d). In addition, the polyglutamate chain of Klf4 could be removed by CCP1 or CCP6 with an in vitro deglutamylation assay (Supplementary Figure 4e).

We next monitored the colocalization of Klf4 and GT335 in the process of reprogramming with immunofluorescence staining. Colocalization of GT335 with Klf4 staining was observed in the nucleus of MEFs after OSKM induction (Supplementary Figure 4f). Consistently, $Ccp1^{-/-}$ or $Ccp6^{-/-}$ MEFs with OSKM transduction showed strong glutamylation signals (Fig. 3f and

Supplementary Figure 4g). More importantly, Klf4 polyglutamy-lation also appeared in the nucleus of early embryos and enhanced after the 4- to 8-cell stage (Supplementary Figure 4h). As expected, hyperglutamylation of Klf4 also appeared in $Ccp6$-deficient embryos (Fig. 4g). Consistently, Klf4 polyglutamylation also appeared in mouse ESCs, and declined during the process of retinoic acid (RA)-induced differentiation (Supplementary Figure 4i). We also assessed whether the reprogramming process induced by non-OSKM factors affected endogenous Klf4 polyglutamylation. We transduced factors Sall4, Nanog, Esrrb, and Lin28 (termed SNEL) into $Ccp6^{-/-}$ MEFs for iPSC induction.

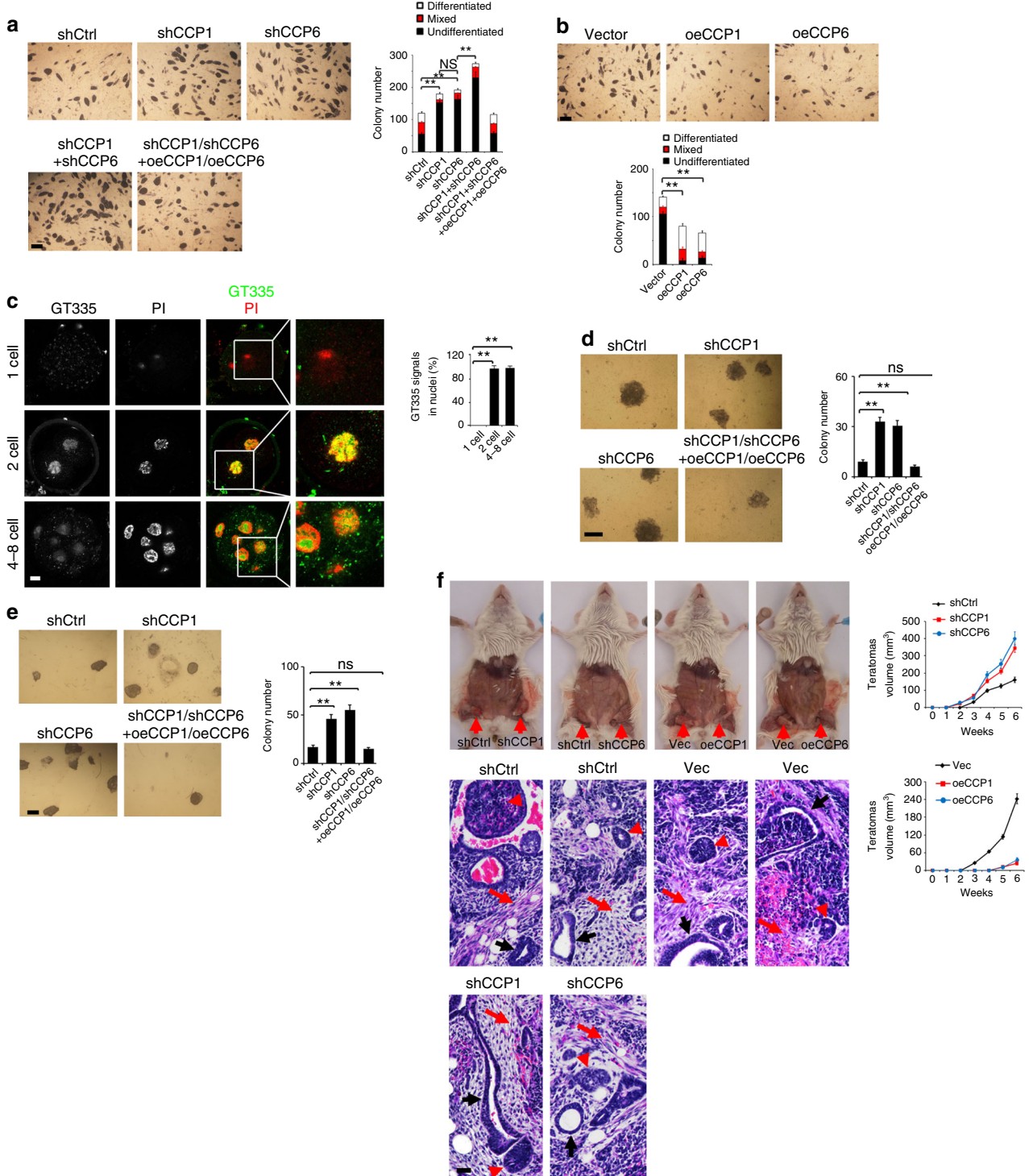

In parallel, SNEL factor transduction in $Ccp6^{-/-}$ MEFs really enhanced hyperglutamylation of Klf4 as well, and consequently generated much more iPSC colonies compared to that of $Ccp6^{+/+}$ MEFs (Supplementary Figure 4j). Taken together, the polyglutamylation modification of Klf4 plays a critical role in the modulation of cell reprogramming and pluripotency.

**Klf4 glutamylation impedes its K48-linked ubiquitination**. We next wanted to explore how Klf4 glutamylation affects its activity. We noticed that Ccp1- and Ccp6-deficient MEFs impeded Klf4 degradation with OSKM transduction (Fig. 4a), whereas Klf4 was rapidly degraded in WT MEFs. In contrast, enforced over-expression of CCP1 or CCP6 in MEFs promoted Klf4 degradation (Supplementary Figure 5a). Restoration of CCP1 or CCP6 in accordance with $Ccp1^{-/-}$ or $Ccp6^{-/-}$ MEFs in turn reduced Klf4 protein levels comparable to that of $Ccp1^{+/+}$ or $Ccp6^{+/+}$ cells (Supplementary Figure 5a). These results indicate that deficiency of CCP1 or CCP6 sustains Klf4 protein stability.

Klf4 binds to the E3 ligase component βTrCP1 or βTrCP2 that catalyzes its ubiquitination leading to proteasome-mediated degradation[16]. Consistently, CCP6 deficiency remarkably declined Klf4 ubiquitination signals during iPSC induction (Fig. 4b). As expected, Klf4 ubiquitination was K48-linked but not K63-linked through K48R- and K63R-ubiquitin transfection assays (Fig. 4c). Notably, TTLL4 overexpression reduced Klf4 ubiquitination that suppressed Klf4 degradation (Fig. 4c). Furthermore, TTLL4-mediated glutamylation of Klf4 competitively inhibited its binding with the E3 ligase component βTrCP1 (Fig. 4d). Accordingly, Klf4 was ubiquitinated by the βTrCP1/ Skp1/Cullin1/Rbx1 E3 ligase complex via an in vitro ubiquitination reconstitution assay (Fig. 4e). TTLL4-mediated glutamylation of Klf4 indeed repressed its ubiquitination. MLN4924 is an inhibitor that inactivates the SCF (Skp1, Cullins, and F-box proteins) complex E3 ubiquitin ligase activity[44]. As expected, MLN4924 treatment augmented Klf4 stability in OSKM-transduced MEFs and consequently promoted iPSC formation efficiency (Supplementary Figure 5b,c). Consistently, Klf4 depletion substantially reduced the gene expression of pluripotency factors (Supplementary Figure 5d, e). In addition, Klf4 occupied the promoter regions of these pluripotency factors (Supplementary Figure 5f).

We generated TTLL4 knockout (KO) mice via a CRISPR–Cas9 approach[45]. An 8-bp deletion of exon 3 led to frameshift mutation and full deletion of TTLL4 in mice (Supplementary Figure 5g, h). We found that with OSKM induction, TTLL4 KO dramatically augmented Klf4 ubiquitination and facilitated Klf4 degradation (Fig. 4f,g). In parallel, TTLL4 deficiency caused hypoglutamylation of Klf4 that resulted in its degradation in 4–8-cell-stage embryos and impaired early embryonic development (Fig. 4h). Finally, TTLL4 KO MEFs with MLN4924 treatment

could restore Klf4 protein levels during OSKM transduction (Supplementary Figure 5i). The specificities of antibodies against CCP1, CCP6, TTLL1, or TTLL4 that we used were verified (Supplementary Figure 5j). Overall, TTLL4-mediated glutamylation of Klf4 impedes its K48-linked ubiquitination that sustains Klf4 stability.

**A Klf4 glutamylation mutant reduces reprogramming frequency**. To further confirm whether Klf4 polyglutamylation regulates reprogramming and pluripotency maintenance, we overexpressed Klf4-wt, Klf4-E381A, or Klf4-K232R (a stable form mutant[46]) into Klf4-silenced MEFs with OSKM transduction. We found that Klf4-E381A mutant abrogated polyglutamylation signals, iPSC formation, and pluripotent gene expression (Fig. 5a and Supplementary Figure 5k, l). By contrast, Klf4-K232R restoration achieved opposite results vs. the Klf4-E381A restoration. Consistently, Klf4-E381A restoration in mouse ESC D3 cells exhibited decreased colony numbers with AP-positive staining and downregulated pluripotent gene expression, while Klf4-K232R restoration in D3 cells displayed increased numbers of AP-positive colonies and upregulated pluripotent gene expression compared with Klf4-wt restoration (Fig. 5b and Supplementary Figure 5m). Furthermore, Klf4-E381A restoration enhanced differentiation gene expression, whereas Klf4-K232R restoration suppressed differentiation gene expression (Supplementary Figure 5n). These data indicate that Klf4 polyglutamylation is required for cell reprogramming and pluripotency maintenance.

Nanog and Esrrb are downstream target genes of Klf4 during the iPSC reprogramming[47,48]. DNase I digestion assay showed that CCP6 deficiency augmented chromatin accessibility to DNase I digestion at the promoters of Nanog and Esrrb (Fig. 5c), indicating an open status of chromatin region of these genes. Rescue of Klf4-E381A into Klf4-silenced MEFs reduced DNase I accessibility of these two gene promoters (Fig. 5c). By contrast, restoration of Klf4-K232R into Klf4-silenced MEFs increased DNase I accessibility of these two promoters. These results suggest that Klf4 stabilization by polyglutamylation initiates transcriptional activation of its downstream target genes.

Actually, Klf4-wt glutamylation was catalyzed by TTLL4, which remarkably reduced its ubiquitination modification (Fig. 5d). By contrast, Klf4-E381A overexpression abrogated the polyglutamylation of Klf4 and augmented its ubiquitination signals (Fig. 5d). Expectedly, Klf4-K232R overexpression declined its ubiquitination modification (Fig. 5d). Similar results were achieved by treatment with the counterparts of TTLL1 over-expression (Supplementary Figure 5o). We next wanted to exclude the possibility that mutants of Klf4 could affect its target gene expression by changing their DNA-binding capacities.

**Fig. 2** Glutamylation is required for the maintenance of mouse and human pluripotency. **a** Mouse ESC D3 cells were transfected with scrambled shRNA (shCtrl), shCCP1, shCCP6 or CCP1 and CCP6 overexpression (oe) plasmids as indicated and cultured in mouse ESC media. After 5 days, pluripotency was analyzed by AP staining. Colony numbers for undifferentiated, mixed or differentiated clones were calculated as means ± S.D. **P < 0.01. Scale bar, 100 μm. n = 4. **b** D3 cells were transfected with the indicated plasmids and assessed AP staining as in **a**. **P < 0.01. Scale bar, 50 μm. n = 6. **c** Embryos at the indicated stages were isolated and stained with GT335 antibody and PI, and visualized by confocal microscopy. Scale bar, 20 μm. Percentages of cells with GT335 signal localized in the nucleus were counted as means ± S.D. **P < 0.01. For 1-cell-stage embryos, n = 48. For 2-cell-stage embryos, n = 56. For 4- to 8-cell stage embryos, n = 62. **d** Human ESC H9 cells were infected with lentivirus expressing the indicated shRNAs and cultured in human ESC media for 3 weeks. Pluripotency was analyzed by AP staining. Scale bar, 100 μm. AP-positive colony numbers per well were calculated as means ± S.D. **P < 0.01. n = 5. **e** Depletion of CCP1 or CCP6 in human iPSCs increases AP+ colony formation. Human iPSCs were infected with lentivirus expressing the indicated shRNAs and cultured in hiPSC media for 3 weeks, followed by AP staining. Scale bar, 50 μm. AP-positive colony numbers per well were calculated as means ± S.D. **P < 0.01. n = 5. **f** CCP1 or CCP6 depletion in human iPSCs promotes teratomas formation. Human iPSCs infected with the indicated lentivirus were injected subcutaneously into NOD/SCID mice. Six weeks later, teratomas were collected for H&E staining. Red arrow denotes teratomas in upper panels. Black arrow, endoderm; red arrow, mesoderm; red arrowhead, ectoderm. Scale bar, 200 μm. Total 6 teratomas from 6 mice were analyzed pre condition (n = 6). The teratomas were from 2 iPSC cell lines. Student's t test was used as statistical analysis. ns no significance

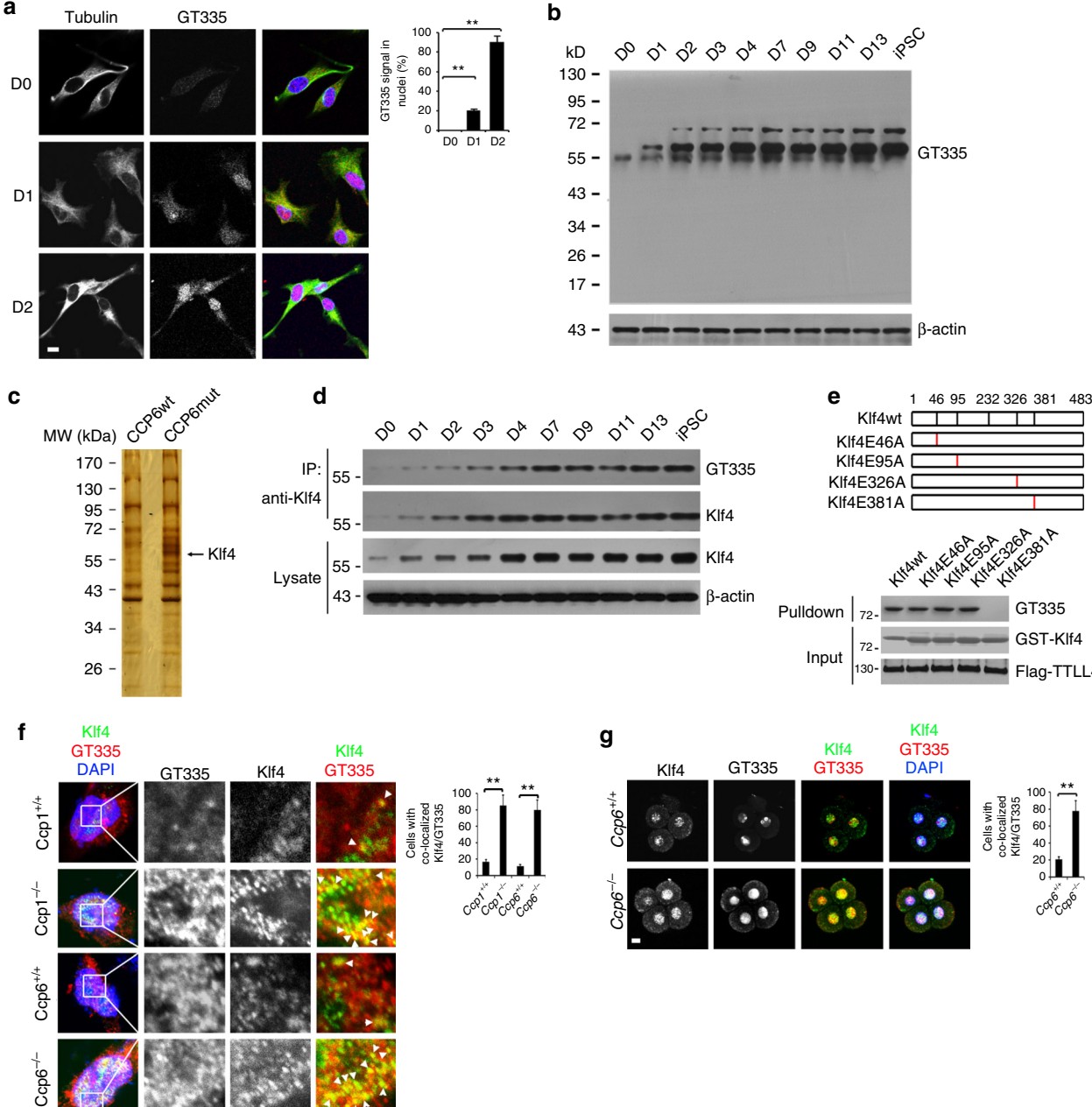

**Fig. 3** Klf4 undergoes polyglutamylation in cell reprogramming and early embryogenesis. **a** MEFs were transduced with STEMCCA lentivirus for the indicated days and immunostained with anti-tubulin and GT335 antibodies. Nuclei were counterstained by DAPI. Scale bar, 10 μm. Percentages of cells with GT335 signal localized in nuclei were counted as means ± S.D. **P < 0.01. More than 100 typical cells were observed and calculated. n = 134 for D0, n = 122 for D1, n = 151 for D2. **b** MEFs lysates induced by OSKM for the indicated days were analyzed by immunoblotting with GT335 antibody. **c** Recombinant CCP6-wt and inactive CCP6 mutant (CCP6-mut) were immobilized to Affi-gel10 resin. Lysates of OSKM-induced MEFs were added for affinity chromatography. The eluted fractions were visualized by SDS-PAGE followed by silver staining. The differential band around 60 kD appeared in CCP6-mut gel was cut for mass spectrometry and identified as Klf4. **d** MEFs Lysates induced by OSKM for the indicated days were immunoprecipitated with anti-Klf4 antibody and analyzed by immunoblotting with GT335. **e** Schematic representation of four putative glutamate-rich glutamate-rich regions in Klf4 (left panel). GST-Klf4wt and indicated mutant proteins were incubated with Flag-TTLL4. GST-Klf4 proteins were pulled down by Glutathione Sepharose 4B beads, followed by immunoblotting. **f** *CCP1-* and *CCP6*-deficient MEFs were transduced with OSKM and stained with anti-Klf4 antibody. Nuclei were stained by DAPI. Scale bar, 10 μm. Percentages of cells with Klf4/GT335 colocalization in nuclei were counted as means ± S.D. **P < 0.01. 102 *Ccp1*+/+ MEF cells, 112 *Ccp1*−/− MEF cells, 104 *Ccp6*+/+ MEF cells and 117 *Ccp6*−/− MEF cells were observed. n = 4. **g** *CCP6*-deficient embryos were isolated and immunostained with anti-Klf4 and GT335 antibodies. Percentages of cells with Klf4/GT335 colocalization in nuclei were counted as means ± S.D. **P < 0.01.110 *Ccp6*+/+ embryos and 116 *Ccp6*−/− embryos were observed. Scale bar, 20 μm. Student's *t* test was used as statistical analysis. The data represent four independent experiments. MW molecular weight marker

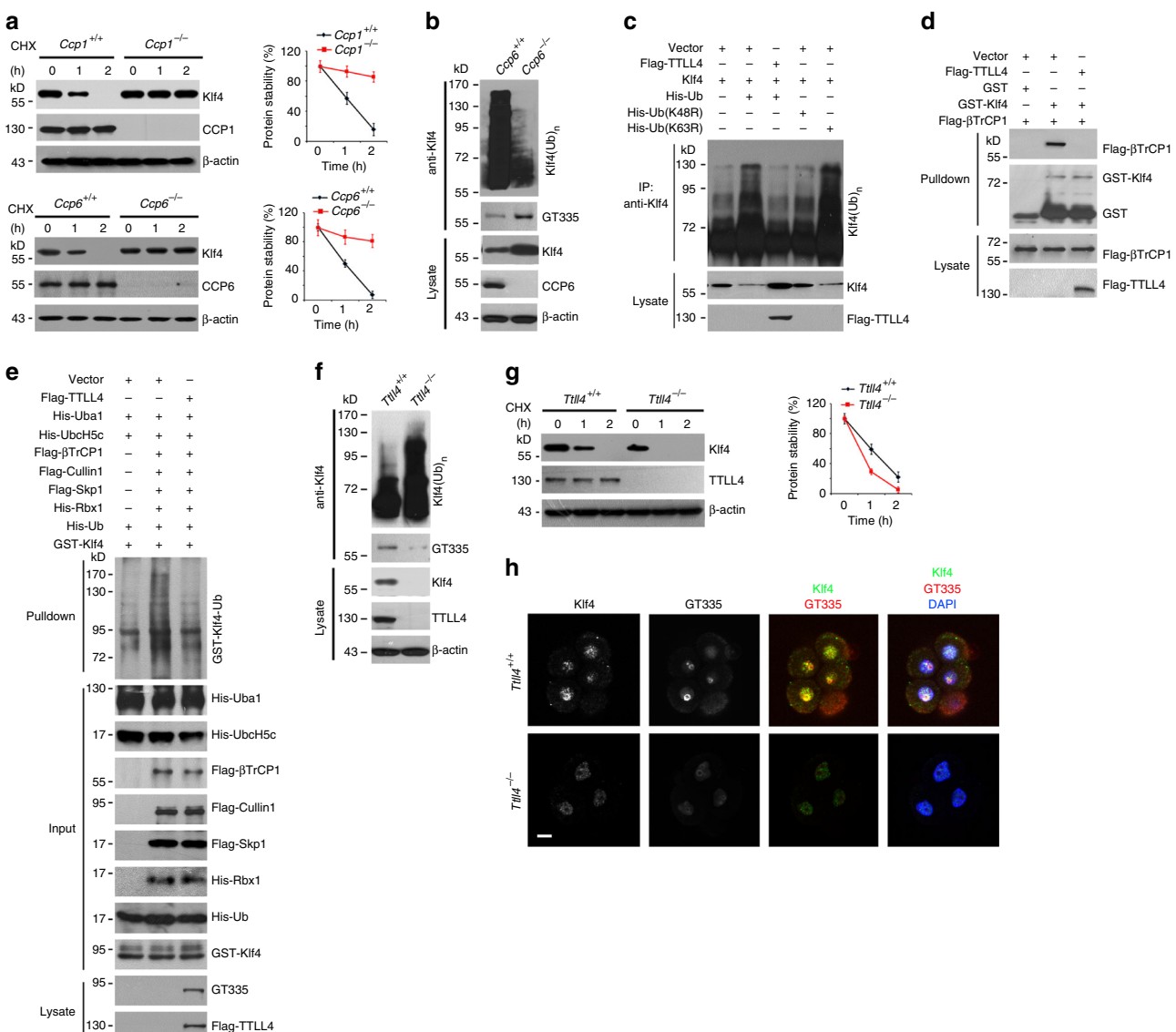

**Fig. 4** Klf4 glutamylation impedes K48-linked ubiquitination to sustain its stability. **a** *Ccp1*- and *Ccp6*-deficient MEFs were treated with cycloheximide (CHX) (20 μg/ml) for the indicated time and followed by immunoblotting. Percentage of residual protein levels was counted as means ± S.D. **b** *Ccp6+/+* or *Ccp6−/−* MEFs were transfected by OSKM for 2 days and treated with MG132 (10 μM) for 4 h. Cells were collected and immunoprecipitated with anti-Klf4 antibody followed by detection with anti-poly-Ub antibody. **c** Klf4 ubiquitination is K48-linked but not K63-linked. pMX5-Klf4 together with pCDNA4-MycHis-Ubiquitin (Ub) or Ub(K48R), Ub(K63R) mutant were transfected into 293T cells for 48 h and analyzed as in **b**. **d** rGST-Klf4 was pre-incubated with lysates from 293 T cells transfected with Flag-TTLL4 at 37 °C for 2 h, followed by GST pulldown with Flag-βTrCP1 and immunoblotting with indicated antibodies. **e** rGST-Klf4 was pre-incubated with lysates from 293T cells transfected with Flag-TTLL4 at 37 °C for 2 h. Klf4 protein was then incubated with recombinant active ERK1 (Millipore) and 2 mM ATP at 30 °C for 30 min. His-tagged E1 (Uba1), E2 (UbcH5c), ubiquitin, Rbx, Flag-tagged βTrCP1, Skp1, Cullin1 and GST-tagged Klf4 were incubated for the in vitro ubiquitination reconstitution assay at 30 °C for 2 h. Ubiquitinated Klf4 was probed by anti-GST antibody. **f** TTLL4 deficiency promotes the ubiquitination of Klf4 in OSKM-induced MEFs. *Ttll4+/+* or *Ttll4−/−* MEFs was transfected by OSKM for 2 days and assessed for ubiquitination as described in **b**. **g** *Ttll4*-deficient MEFs were treated with cycloheximide (CHX) (20 μg/ml) for the indicated time, followed by immunoblotting as described in **a**. **h** *Ttll4*-deficient 4- to 8-cell stage embryos were isolated and immunostained with anti-Klf4 and GT335 antibodies. Scale bar, 20 μm.For *Ttll4+/+* embryos, $n = 67$. For *Ttll4−/−* embryos, $n = 55$. All data are representative of four independent experiments

Binding capacity of *Nanog* promoter with Klf4-wt, Klf4-E381A, and Klf4-K232R was measured by an electrical mobility shift assay (EMSA). We observed that Klf4-E381A and Klf4-K232R mutants showed comparable binding affinities to Klf4-wt protein (Fig. 5e). Unlabeled *Nanog* probes could competitively bind to the *Nanog* promoter. These data suggest that Klf4 mutation per se has no direct activation effect on *Nanog* gene. Therefore, polyglutamylation-mediated Klf4 stability directly regulates the transcriptional activation of its downstream target genes.

To further verify the role of Klf4 polyglutamylation in the reprogramming and pluripotency regulation in vivo, we generated Klf4-E381A mutant knockin (*Klf4E381A* KI) mice by CRISPR/Cas9 editing technology. As expected, MEFs isolated from *Klf4E381A* KI mice really abrogated iPSC formation by in vitro OSKM transduction (Fig. 6a). Of note, *Klf4E381A* KI mice impaired blastocyst development at E3.5 (Fig. 6b). In addition, Klf4 with cell fate markers (Nanog for epiblast, Gata6 for primitive endoderm, and Cdx2 for trophectoderm) in early

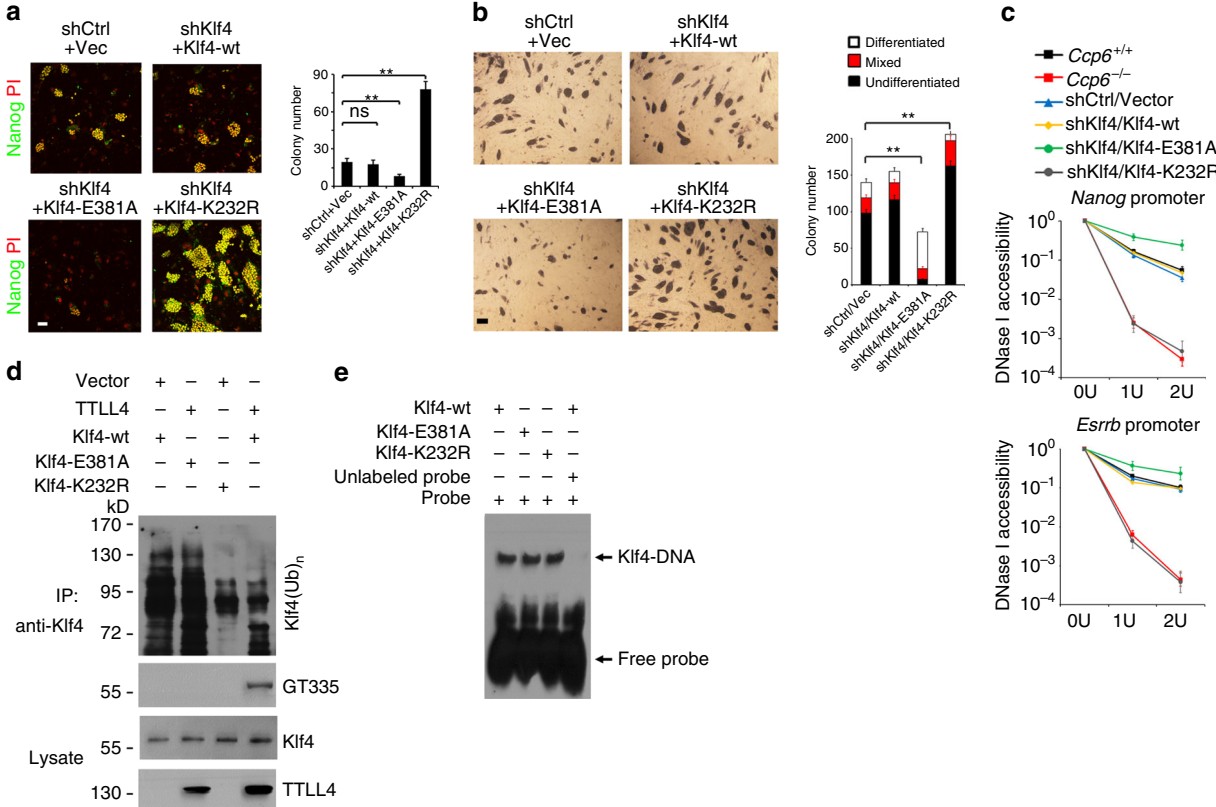

**Fig. 5** A glutamylation-defective mutant of Klf4 reduces reprogramming efficiency. **a** Klf4-wt, Klf4-E381A, or Klf4-K232R were overexpressed in Klf4-silenced 4F2A MEFs and treated with dox and cultured in ESC media for 3 weeks. Scale bar, 50 μm. Reprogramming efficiency was assayed by Nanog staining after dox removal. Nanog-positive colony numbers per $10^4$ cells were calculated and shown as means ± S.D. **$P < 0.01$,. $n = 4$. **b** D3 ESC cells were transfected with indicated plasmids, followed by AP staining. Scale bar, 50 μm. AP-positive colony numbers per well were calculated as means ± S.D. **$P < 0.01$. $n = 5$. **c** SSEA-1[+] ES cells were isolated and an equal amount of cells were lyzed for nuclei extraction, followed by DNase I digestion. Total DNA was extracted and quantitated by qPCR with Nanog or Esrrb promoter-specific primers. $n = 5$. **d** Indicated plasmids along with TTLL4 were overexpressed into Klf4-silenced MEFs after OSKM transfection for 2 days, and cells were collected and immunoprecipitated with anti-Klf4 antibody followed by detection with anti-poly-Ub antibody. **e** Indicated proteins were incubated with labeled and unlabeled probes against Nanog promoter, followed by EMSA assay. Student's t test was used as statistical analysis. ns, no significance

embryos from *Klf4^E381A* KI mice was substantially reduced and consequently underwent apoptosis (Fig. 6c,d). Finally, *Klf4^E381A* homozygous pups were embryonically lethal (Fig. 6e). Functions of Klf2 and Klf5 are redundant with Klf4 in ES cells[35] and iPSCs[49]. Of interest, we found that Klf4-E381A was dimerized with Klf5 in ESCs, while Klf4-wt protein did not (Supplementary Figure 5p). We propose that the dimerization of Klf4-E381A with Klf5 could inactivate the Klf5 function to cause preimplantation lethality in *Klf4^E381A* KI mice. It has been reported that a glutamylation site can shift to the next available glutamate residue when the main modification site is mutated in tubulin[50]. However, Klf4 E381A mutant knockin abrogated the Klf4 function in vivo, excluding the possibility of modification site shift for Klf4 glutamylation. Taken together, Klf4 polyglutamylation at Glu381 is required for cell reprogramming and early embryonic development.

**TTLL1 or TTLL4 deletion impairs embryonic development**. To further explore the physiological role of TTLL1 and TTLL4 in the process of reprogramming, TTLL1 and TTLL4 were silenced in MEFs for iPSC formation assays. Depletion of TTLL1 or TTLL4 dramatically reduced iPSC colony numbers as well as expression levels of pluripotent genes (Fig. 7a and Supplementary Figure 6a, b). Double knockdown of TTLL1 and TTLL4 displayed a synergistic

inhibitory effect. Overexpression of TTLL1 or TTLL4 in accordance with silenced MEFs restored iPSC colonies comparable to shCtrl-treated MEFs (Fig. 7a). TTLL1 or TTLL4 depletion in OSKM-transduced MEFs substantially declined Klf4 glutamylation signals (Supplementary Figure 6c). By contrast, overexpression of TTLL1 or TTLL4 in accordance with silenced MEFs rescued Klf4 glutamylation signals (Supplementary Figure 6c), which in turn promoted iPSC colony formation efficiency (Supplementary Figure 6d, e). Consequently, TTLL1 or TTLL4 depletion promoted Klf4 degradation (Fig. 7b). In contrast, overexpression of TTLL1 and TTLL4 sustained Klf4 stability. Finally, *Ttll4*-deficient MEFs impaired iPSC formation (Supplementary Figure 6f). These data indicate that TTLL1- or TTLL4-mediated Klf4 glutamylation is required for somatic cell reprogramming.

We next analyzed the role of TTLL1 and TTLL4 in ESC pluripotency maintenance. We silenced TTLL1 or TTLL4 expression in mouse ESC D3 cells. As expected, TTLL1- or TTLL4-depleted ESCs exhibited decreased numbers of colonies with AP-positive staining and downregulated pluripotent gene expression compared with shCtrl-treated cells (Fig. 7c,d). Double knockdown of TTLL1 with TTLL4 synergetically lowered AP-positive colony numbers and pluripotent gene expression in ESCs. Additionally, restoration of TTLL1 or TTLL4 into its corresponding TTLL1- and TTLL4-silenced cells rescued ESC pluripotency and pluripotent gene expression (Fig. 7c,d). By

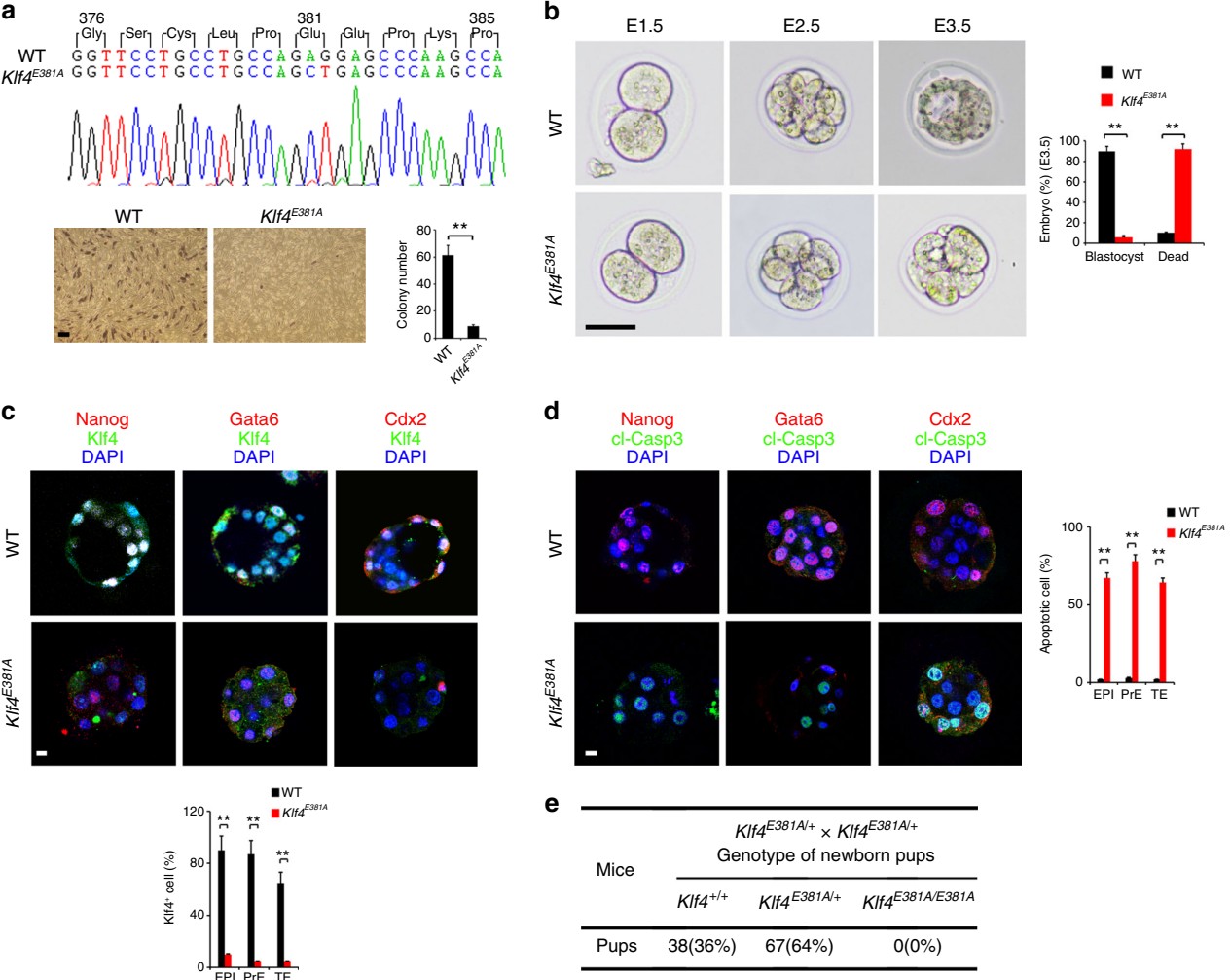

**Fig. 6** A glutamylation-defective mutant of Klf4 impairs ESC pluripotency. **a** *Klf4*$^{E381A}$ knockin mice were generated by a CRISPR/Cas9 approach. MEFs were isolated and induced for iPS formation as in Fig. 5a. Scale bar, 100 μm. AP-positive colony numbers per 10$^4$ cells were calculated and shown as means ± S.D. **$P < 0.01$, $n = 4$. **b** Fertilized eggs at E3.5 embryonic stage were isolated from pregnant mice after heterozygotes crossing. Embryo genotype was confirmed by both GT335 immunostaining and PCR of genomic DNA. Scale bar, 100 μm. Distribution of embryonic stage at E3.5 was counted as means ± S.D, **$P < 0.01$. For WT embryos, $n = 101$. For *Klf4*$^{E381A}$ embryos, $n = 87$. **c** Indicated preimplantation embryos from WT and *Klf4*$^{E381A}$ knockin mice were stained with cell fate markers (Nanog for epiblast (EPI), Gata6 for primitive endoderm (PrE), and Cdx2 for trophectoderm (TE)) together with Klf4. $n = 34$ for WT embryos and $n = 28$ for *Klf4*$^{E381A}$ embryos (anti-Nanog staining), $n = 41$ for WT embryos and $n = 25$ for *Klf4*$^{E381A}$ embryos (anti-Gata6 staining), and $n = 31$ for WT embryos and $n = 29$ for *Klf4*$^{E381A}$ embryos (anti-Cdx2 staining). **d** Indicated preimplantation embryos were stained with cell fate markers as well as cleaved caspase 3 (cl-Casp3). The percentage of positive cell in each germ layer was counted as means ± S.D, **$P < 0.01$. Scale bar, 20 μm. $n = 40$ for WT embryos and $n = 31$ for *Klf4*$^{E381A}$ embryos (anti-Nanog staining), $n = 37$ for WT embryos and $n = 30$ for *Klf4*$^{E381A}$ embryos (anti-Gata6 staining), and $n = 37$ for WT embryos and $n = 25$ for *Klf4*$^{E381A}$ embryos (anti-Cdx2 staining). **e** *Klf4*$^{E381A}$ pups were genotyped after heterozygotes crossing. Student's *t* test was used as statistical analysis

contrast, enforced overexpression of TTLL1 or TTLL4 increased ESC self-renewal as well as upregulated pluripotent gene expression (Fig. 7e,f). Similar results were obtained in TTLL1- or TTLL4-depleted human ESC H9 cells (Fig. 7g,h).

To further test the role of TTLL1 and TTLL4 in early embryogenesis, we also generated *TTLL1* knockout (KO) mice via a CRISPR–Cas9 approach[45] (Supplementary Figure 7a, b). We isolated fertilized eggs from WT, *Ttll1*$^{-/-}$, and *Ttll4*$^{-/-}$ pregnant mice to assess *ex vivo* development of embryos. Both TTLL1 and TTLL4 were deleted in embryos by immunofluorescence staining (Fig. 8a). TTLL1 and TTLL4 deficiency suppressed blastocyst development (Fig. 8b), leading to decreased blastocoel areas, disordered lineage marker expression, and increased apoptotic cells of early embryos (Supplementary Figure 7c-f). Finally, both

*Ttll4*-deficient pups and *Ttll1*-deficient pups were born in extremely lower numbers than those that would be predicted by Mendelian inheritance after heterozygotes crossing (Fig. 8c and Supplementary Figure 7g). Altogether, TTLL1 and TTLL4 play a critical role in the regulation of iPSC formation and early embryonic development.

## Discussion

Transcription factors-mediated reprogramming was sufficient to reset terminally differentiated cells into induced pluripotent stem cells (iPSCs)[2,32,51]. The generation of iPSCs avoids ethical issues by using embryonic stem cells. Additionally, iPSCs can be established from patient's tissues, which makes them more

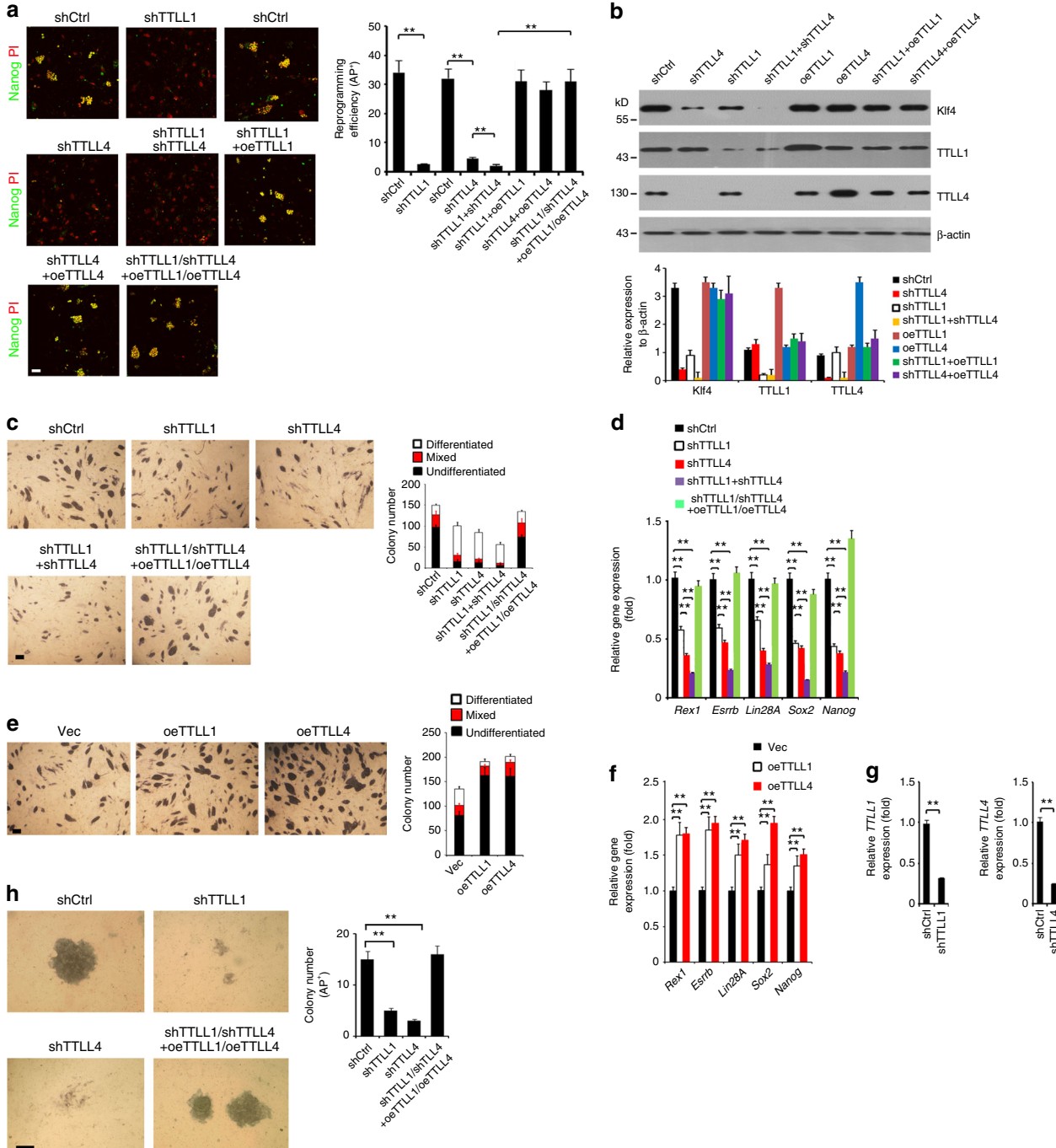

**Fig. 7** Deletion of TTLL1 or TTLL4 impairs iPSC induction and ESC pluripotency. **a** Depletion of TTLL1 or TTLL4 impairs iPSC formation. TTLL1 or TTLL4 was silenced in 4F2A MEFs as well as rescued with TTLL1 or TTLL4 into their silenced MEFs, followed by dox treatment for iPSC formation and stained with anti-Nanog antibody as in Fig. 1e. Scale bar, 50 μm. Nanog-positive colony numbers per $10^4$ cells were calculated and shown as means ± S.D. **P < 0.01. n = 5. **b** Immunoblotting of Klf4 in above-treated MEFs as in **a**. Fold changes of relative expression of indicated proteins compared with β-actin were caculated as means ± S.D. The data represent four independent experiments. **c** TTLL1 or TTLL4 depletion enhances the pluripotency of mouse ESCs. D3 cells were transfected with scrambled shRNA (shCtrl), shTTLL1, shTTLL4, or TTLL1 and TTLL4 overexpression (oe) plasmids and cultured in mouse ESC media. After 5 days, pluripotency was analyzed by AP staining. Colony numbers for undifferentiated, mixed, or differentiated clones were calculated as means ± S.D. n = 5. Scale bar, 100 μm. **d** D3 cells were transfected with indicated plasmids. mRNA levels of the indicated genes were analyzed by real-time qPCR. Relative gene expression fold changes were counted as means ± S.D. **P < 0.01. n = 5. Primer pairs are shown in Supplementary Table 1. **e** Indicated plasmids were transfected into mouse D3 cells as in **c**. Colony numbers for undifferentiated, mixed, or differentiated clones were calculated as means ± S. D. n = 5. Scale bar, 100 μm. **f** mRNAs levels of the indicated genes were analyzed by real-time qPCR as in **d**. Relative gene expression fold changes were counted as means ± S.D. **P < 0.01. n = 5. **g** TTLL1 or TTLL4 depletion in human ESC H9 cells was confirmed by real-time qPCR. n = 5. **h** TTLL1 or TTLL4 depletion in human ESC H9 cells increases AP⁺ colony formation. Human H9 cells were infected with lentivirus expressing the indicated shRNAs and cultured in human ES media for 3 weeks. ESC pluripotency was analyzed by AP staining. AP⁺ colony numbers were calculated as means ± S.D. n = 4. Scale bar, 100 μm. Student's t test was used as statistical analysis. oe overexpression

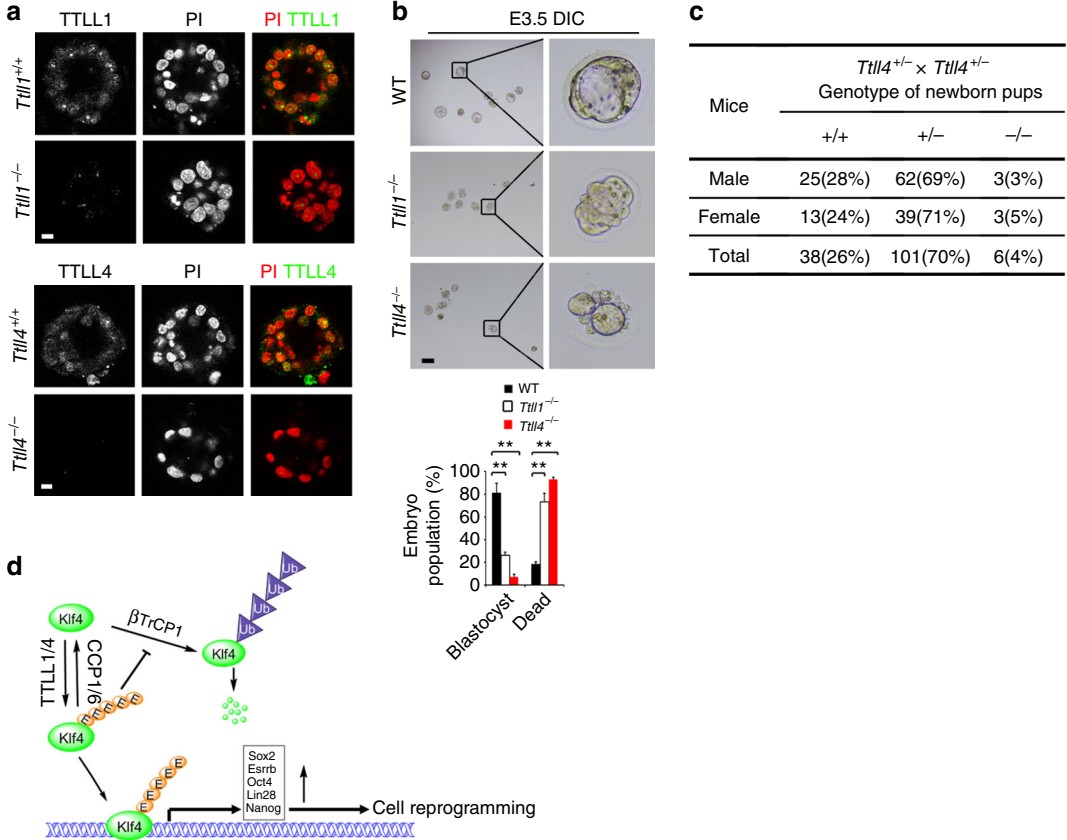

**Fig. 8** Deletion of TTLL1 and TTLL4 impairs embryonic development. **a** TTLL1 and TTLL4 deficiency in blastomere was confirmed by immunofluorescence staining. Scale bar, 20 μm. **b** Fertilized eggs were isolated from WT, *Ttll1*⁻/⁻, and *Ttll4*⁻/⁻ pregnant mice and assessed as in Fig. 1f. Distribution of embryonic stage at E3.5 was counted as means ± S.D (right panel), **P < 0.01. Scale bar, 100 μm. For WT embryos, n = 115. For *Ttll1*⁻/⁻ embryos, n = 101. For *Ttll4*⁻/⁻ embryos, n = 108. **c** *Ttll4*-deficient pups were genotyped after heterozygotes crossing. Student's *t* test was used as statistical analysis. Data are representative of four independent experiments. **d** A working model represents Klf4 polyglutamylation in the regulation of cell reprogramming and early embryogenesis. During the process of reprogramming and pluripotency maintenance, the core pluripotency factor Klf4 is polyglutamylated by TTLL4 or TTLL1 that impedes its K48-linked ubiquitination to maintain Klf4 stability. Deletion of TTLL4 or TTLL1 impairs cell reprogramming and early embryogenesis. Klf4 glutamylation is hydrolyzed by CCP1 or CCP6. CCP1 or CCP6 deficiency substantially promotes iPSC generation and sustains ESC pluripotency. Glutamylation of Klf4 is required for activation of its downstream pluripotency genes leading to somatic reprogramming, which maintains ESC pluripotency as well as drives embryogenesis

reliable and beneficial for treatment of human-related diseases[52]. In this study, we show that CCP1 or CCP6 deficiency substantially promotes iPSC generation and sustains ESC pluripotency. During the process of reprogramming, the core pluripotency factor Klf4 is polyglutamylated by TTLL4 or TTLL1 that impedes its K48-linked ubiquitination to maintain Klf4 stability (Fig. 8d). TTLL1- or TTLL4-mediated polyglutamylation of Klf4 is required for activation of its downstream pluripotency genes leading to somatic reprogramming, which maintains ESC pluripotency as well as drives embryogenesis. Deletion of TTLL4 or TTLL1 impairs cell reprogramming and early embryogenesis. Therefore, Klf4 polyglutamylation plays a critical role in the regulation of cell reprogramming and pluripotency maintenance.

Protein levels of Oct4 and Sox2 are precisely modulated in order to sustain self-renewal and pluripotency of ESCs[53,54]. A moderate increase of Sox2 causes differentiation of ESCs primarily into neural ectodermal cells, mesodermal, and trophectoderm-like cells[53], while decreased expression of Sox2 induces differentiation of ESCs into trophectoderm-like cells[55]. Similarly, Klf4 activity is also tightly regulated by PTMs. For example, phosphorylation of Klf4 by ERKs at Ser123 leads to inhibition of its transcriptional activity[16]. In addition, Klf4 also

undergoes sumoylation that enhances Klf4 transactivation activity[56]. However, it is unknown whether core transcription factors undergo glutamylation in cell reprogramming and ESC pluripotency. The well-known substrates of polyglutamylation are tubulins and nucleosome assembly proteins[43,57,58]. A recent study delineates a structural microtubule recognition basis by catalysis with TTLL7[19]. TTLLs have different expression patterns in diverse tissues and their functions are not entirely redundant[59]. We recently reported that TTLL4 and TTLL6 are most highly expressed in megakaryocytes[29], both of which catalyze polyglutamylation of Mad2 to modulate megakaryocyte maturation. In this study, we demonstrate that TTLL1 and TTLL4 are constitutively elevated in ESCs and early embryos, and dramatically upregulated in iPSCs during cell reprogramming. Both TTLL1 and TTLL4 can catalyze polyglutamylation of Klf4, which impedes its K48-linked ubiquitination to maintain Klf4 stability. We showed that both *Ttll4*-deficient pups and *Ttll1*-deficient pups were born in extremely lower numbers than those that would be predicted by Mendelian inheritance after heterozygotes crossing. These observations suggest that TTLL1 and TTLL4 might synergistically modify the same glutamate on Klf4 to regulate early embryonic development. However, the different biochemical activities of TTLL family enzymes in the

regulation of cell reprogramming still need to be further investigated.

Protein glutamylation is a reversible modification, whose deglutamylation is hydrolyzed by a family of cytosolic carboxypeptidases (CCPs)[23,60]. CCP family members harbor enzymatic specificities to carry out deglutamylation. Of these CCP members, CCP1, CCP4, and CCP6 remove the shortening of penultimate polyglutamate chains of α-tubulin, while CCP5 specifically hydrolyzes the branching site glutamate[23]. We previously showed that Mad2 polyglutamylation can be hydrolyzed by CCP6 but not CCP1[23]. We provide the example that CCP1 and CCP6 harbor nonredundant roles in physiological conditions. Consequently, deficiency of CCP1 or CCP6 exhibits different abnormalities[27,29]. Herein, we show that both CCP1 and CCP6 are highly expressed in differentiated cells and remarkably declined in ESCs and iPSCs. Both CCP1 and CCP6 can hydrolyze the glutamylation of Klf4 during cell reprogramming and ESC maintenance. CCP1 and CCP6 could synergistically shorten longer glutamate chains of glutamyated Klf4, while CCP5 may be needed to deglutamylate Klf4 at a branching glutamate site. Alternatively, CCP1 and CCP6 could also remove a branched glutamate of glutamyated Klf4, whose biochemical activities may be different from tubulins[23].

Glutamylation is highly conserved in all metazoans and protists, performing the assembly and function of cilia and flagella[58]. For example, TTLL7, the most abundantly expressed in the mammalian nervous system, is conserved from acorn worm to primates, where it modulates neurite outgrowth and localization of dendritic MAPs. In addition, CCP and TTLL members exert their functions by enzymatic activities against their respective substrates, and their enzymatic activities are also modulated by post-translational modifications and other regulatory factors. Here, we show that Klf4 polyglutamylation mediated by TTLL1 and TTLL4 enhances mouse and human reprogramming and pluripotency. In parallel, the CCP inhibitor phenanthroline dramatically promotes somatic cell reprogramming and embryonic development. Thereby, we strongly believe that it is necessary to develop specific inhibitors or agonists for each individual polyglutamylase and cytosolic carboxypeptidase. Manipulating polyglutamylation profiles by using these compounds, we may potentially improve iPSC efficiency for future clinical applications. In sum, Klf4 glutamylation plays a critical role in the regulation of cell reprogramming and pluripotency maintenance. Our findings provide mechanistic insights into how glutamylation modulates cell fate determination.

## Methods

**Antibodies and reagents.** Antibodies against Myc-tag (sc-40), TTLL1 (sc-86929), CCP6 (sc-138893), SSEA-1(sc-21702), and GST-tag (sc-138) were purchased from Santa Cruz Biotechnology (Santa Cruz, USA). Antibodies against Flag-tag (M2, F3165), β-actin (A-5316), His-tag (H1029), glutamylated tubulin (B3), and GFP-tag (G-1544) were from Sigma-Aldrich (St. Louis, USA). GT335 (anti-glutamylation) antibody was from AdipoGen (San Diego, USA). The antibody against TTLL4 (PAB2002) was from Abnova (Walnut, USA). Antibodies against ubiquitin (K48 linked) (ab140601) and Gata6 (ab22600) were purchased from Abcam (Cambridge, USA). Anti-Klf4 (11880-1-AP) antibody, anti-CCP1(14067-1-AP) antibody, and antibody against Nanog (14295-1-AP) were purchased from ProteinTech (Chicago, USA). The antibody against tubulin (AT819) was from Beyotime (Shanghai, China). The antibody against CDX2 (CDX2-88) was from BioGenex (Fremont, USA). The antibody against cleaved caspase 3 (#9661) was from Cell Signaling Technology (Beverly, USA). Antibodies against Myc-tag, GST-tag, Flag-tag, β-actin, GFP-tag, and His-tag were used in 1:5000 dilutions for western blotting. Antibodies against TTLL1, CCP6, TTLL4, Klf4, ubiquitin (K48 linked), glutamylated tubulin (B3), CCP1, and GT335 were used in 1:500 dilutions for western blotting. Antibodies against SSEA1 and cleaved Caspase3 were used in 1:500 dilutions for flow cytometric staining. Antibodies against ubiquitin (K48 linked) (ab140601) and Gata6, tubulin, CDX2, β-actin, Nanog, and cleaved caspase 3 were used in 1:500 dilutions for immunofluorescence staining. Alexa-405, Alexa-488, Alexa-594, and Alexa-649-labeled secondary antibodies were from Thermo Fisher Scientific (Waltham, USA). Chicago Sky Blue 6B, DAPI (4,6-diamino-2-phenyl indole), cycloheximide (CHX), MG132, phenanthroline, and NEDD8-activating

enzyme inhibitor (MLN4924) were purchased from Sigma-Aldrich. DNaseI was purchased from Roche Molecular Biochemicals (Basel, Switzerland). CoCl$_2$ was from Sinopharm Chemical Reagent Co.,Ltd (Beijing, China).

**Animals.** $Ccp1^{-/-}$ mice ($Pcd$ mice, BALB/cByJ-Agtpbp1$^{pcd-3J}$-J) were obtained from Jackson Lab. $Ccp6^{-/-}$ mice (FVB background) were generated by a transposon piggyBac insertional mutagenesis program (PBMICE). The strategy of breeding was to cross between heterozygotes to generate homozygote knockout mice and littermate control mice as described[29]. Mouse experiments were approved by the Institutional Animal Care and Use Committees at the Institute of Biophysics, Chinese Academy of Sciences. All the mice that we used were from C57BL/6 background, female, and 8–12-weeks old. We were not blinded to the group and did not use randomization in our animal studies. Littermates with the same age and gender for each group were used. We excluded the mice that were 5 g thinner than other littermates before any treatment or analysis. All mice were bred in SPF animal facility. The euthanasia method was cervical dislocation when needed.

**Cell culture.** Human 293 T cells were cultured with DMEM supplemented with 10% FBS and 100 U/ml penicillin and 100 mg/ml streptomycin. Retrovirus- or lentivirus-infecting MEFs were produced in 293 T cells using the standard protocols. Transfection was performed using lipofectin (Invitrogen). A total of 293 T cells were obtained from ATCC and authenticated by PCR. Mycoplasma contamination was tested by PCR. Mouse ESC D3 cells were from the stem cell core facility of Shanghai Institute of Biochemistry and Cell Biology, Chinese Academy of Sciences (Shanghai, China). D3 or R1 ESCs were cultured in DMEM supplemented with 15% FBS (Invitrogen, USA), 2 mM L-glutamine, 0.1 mM 2-mercaptoethanol, 1 mM nucleosides, 0.1 mM nonessential amino acids, and 10$^3$ units/ml mouse leukemia inhibitory factor (LIF) and were grown on B6 mouse MEFs treated with mitomycin C. Human ESC H1 cells were obtained from Wicell Research Institute (Madison, WI). Human ESCs and iPSCs were grown on CF-1 mouse MEFs treated with mitomycin C as feeder cells. Culture medium contained DMEM/F12 medium, 20% knockout serum replacement (KSR) (Invitrogen), and 2 mM L-glutamine, 0.1 mM nonessential amino acids, 0.1 mM 2-mercaptoethanol, and 4 ng/ml hFGF2 (Sigma-Aldrich, USA). Concentrations of 10 μM CoCl$_2$ or 1 μM phenanthroline were used during iPSC induction. A concentration of 1 μM MLN4924 was used for iPSC induction[44]. Embryonic stem cell use was approved by the Institutional Medical Research Ethics Committee at the Institute of Biophysics, Chinese Academy of Sciences.

**Plasmid construction and protein expression.** Mouse CCP6, ubiquitin (Ub), Ub (K48R), and Ub(K63R) were cloned into pCDNA4-MycHis expression vector. CCP6-H230S/E233Q mutant (CCP6mut) and CCP6wt were also subcloned into H-MBP-3C vector and purified using amylose resin (New England BioLabs, Ipswich, USA) according to the manufacturer's instruction. CCP1, TTLL4, Cullin1, and Skp1 were cloned into p3×flag-CMV-9 expression vector. pMX5-Oct4, pMX5-Sox2, pMX5-Klf4, and pMX5-cMyc plasmids were from Dr. Yang Xu's lab (University of California, San Diego, La Jolla, USA). pDEST-Flag-PHF13-βTrCP1 plasmid was from Dr. Degui Chen's lab (Shanghai Institute of Biochemistry and Cell Biology, Chinese Academy of Sciences, Shanghai, China). STEMCCA plasmid was from EMDMillipore. Mouse wild-type Klf4 (Klf4wt), Klf4E46A, Klf4E97A, Klf4K232R, Klf4E329A, and Klf4E384A mutants were generated and cloned into pGEX-6P-1 vector (GE Healthcare), expressed in *E. coli*, and purified using Glutathione Sepharose 4B beads. Uba1, UbcH5c, Rbx, and ubiquitin were subcloned into pET-28a vectors. CCP1, CCP6, TTLL4, TTLL1, Klf4wt, Klf4E384A mutant, and Klf4K232R mutant were also subcloned into pMY-IRES-GFP vector (Cell Biolabs) for enforced expression. RNA interference was designed according to BLOCK-IT RNAi Designer system instructions (Invitrogen). shRNA oligos encoding target mouse sequences against mKlf4 (5′-GACATCGCCGGTTT ATATTGA-3′), mCCP1 (5′-GGAGAACACGAAAGATCTTCA-3′), mCCP6 (5′-C CGAGTCTGGTTCAACTTTAC-3′), mTTLL1(5′-GGATGAGCGTGCAAACC ATTC-3′), mTTLL4 (5′-CCAGCCAGCCTATTTCCTTTG-3′), or their accordingly scrambled sequences were cloned into MSCV-LTRmiR30-PIG vector (LMP, Openbiosystems). shRNA oligos encoding target human sequences against hCCP1 (5′-GCATAGAAACATGCTCATTCG-3′), hCCP6 (5′-GCTGAGGACTTCTCCT ATTCC-3′), or their accordingly scrambled sequences were cloned into pSUPER vector (Oligoengine, Seattle, USA).

**iPSC generation.** Mouse embryonic fibroblasts (MEFs) were obtained at E14.5 of $Ccp1^{+/+}$, $Ccp1^{-/-}$, $Ccp6^{+/+}$, $Ccp6^{-/-}$, $Ttll4^{+/+}$, or $Ttll4^{-/-}$ embryos and supplied with Dulbecco's modified Eagle's medium (DMEM) (Invitrogen) with 15% FBS, 100 μg/ml streptomycin, and 100 U/ml penicillin. MEFs were infected with pMX5-Oct4, pMX5-Sox2, pMX5-Klf4, and pMX5-cMyc-packaged retrovirus mixture and then plated on mitomycin-C-treated feeder cells and cultured in ESC media containing DMEM supplemented with 15% FBS (Invitrogen, USA), 2 mM L-glutamine, 0.1 mM 2-mercaptoethanol, 1 mM nucleosides, 0.1 mM nonessential amino acids, and 10$^3$ units/ml mouse leukemia inhibitory factor (LIF). For higher 4-factor infection, MEFs were infected with STEMCCA lentivirus for reprogramming induction. Reprogramming experiments were also performed using MEFs obtained from E14.5 embryos of *4F2A* mice (Jax stock No. 011004). The 4 Yamanaka factor

expression was induced by doxycycline treatment. After 3 weeks, reprogramming efficiency was assayed by Nanog staining after dox removal.

**Embryo collection and culture**. A total of 5 IU of pregnant mare serum gonadotropin (PMSG) was intraperitoneally injected for superovulation and 5 IU of human chorionic gonadotropin (hCG) was injected 48 h later. Embryos were flushed with M2 medium (Millipore) from oviducts or uteri at the indicated stages and cultured *ex vivo* in KSOM medium (Millipore) supplied with 1 mg/ml BSA at 37 ℃ with 5% $CO_2$. Concentrations of 10 μM $CoCl_2$ or 1 μM phenanthroline were used for embryo culture. For each group, more than 100 typical embryos were observed.

**Generation of knockout and *Klf4^{E381A}* knockin mice**. The genome loci of *Ttll1* and *Ttll4* genes were edited by a CRISPR–Cas9 approach[45]. Briefly, vector pST1374 expressing Cas9 and pUC57kan-T7 expressing sgRNA for *Ttll1* or *Ttll4* gene was constructed as below[45]. Mixtures of Cas9 mRNA (100 ng/μl) and sgRNA (50 ng/μl) were microinjected into the cytoplasm of C57BL/6 fertilized eggs. The injected zygotes were cultured until blastocyst stage and transferred into the uterus of pseudopregnant ICR females. Frameshift mutations were identified by PCR screening and DNA sequencing. The genome loci of *Ttll1* and *Ttll4* genes were edited by a CRISPR–Cas9 approach[45]. For generation of *Klf4^{E381A}* knockin mice, the genome locus of *Klf4* gene was knocked in with *Klf4^{E381A}* mutation via a CRISPR–Cas9 approach. A mixture of Cas9 mRNA, sgRNA, and Klf4-E381A templates was microinjected into the cytoplasm of C57BL/6 fertilized eggs and transferred into the uterus of pseudopregnant ICR females. *Klf4^{E381A}* mutations were identified by PCR screening and DNA sequencing. sgRNA sequences are as follows: Ttll1 (5′-CGATCTGGTCGTCTGACAGCCGG-3′); Ttll4 (5′-TTTGCCTC ACGTTGGTGCGGCGG-3′); and Klf4 E381A (5′-ACCGGGTTCCTGCCT GCCAGAGG-3′). Klf4 template sequence (5′-TAGGTTTTGCCACAGCCTGCAT AGTCACAAGTGTGGGTGGCTGTTCTTTTCCGGGGCCACGACCTTC TTCCCCTCTTTGGCTTGGGCTCAGCTGGCAGGCA-3′).

**In vivo assay of teratomas**. Human iPSCs were generated by transducing OSKM factors into human urothelial cells. Urine samples were recruited from three healthy donors with informed consent based on approval by the Institutional Ethical Committees at the Institute of Biophysics, Chinese Academy of Sciences. *Ccp1^{−/−}* or *Ccp6^{−/−}* mouse iPS cells or human iPS cells infected with shCCP1, shCCP6, oeCCP1, and oeCCP6 lentiviruses were collected, washed twice with PBS, and then subcutaneously injected into the bilateral inguens of male NOD/SCID mice ($2 \times 10^6$ cells per injection). After 4 weeks for mouse iPSCs and 6 weeks for human iPSCs, mice were killed. Tumors were weighed, fixed in 4% paraformaldehyde, and sectioned for staining with hematoxylin and eosin (H&E)[61]. The maximal sizes of tumors permitted were 2500 mm$^3$ according to the ethical approval by the Institutional Animal Care and Use Committees at the Institute of Biophysics, Chinese Academy of Sciences.

**Immunofluorescence assay**. MEFs were fixed with 4% paraformaldehyde (PFA, Sigma-Aldrich) for 20 min after retrovirus infection. Embryos were collected and treated with acidic tyrode buffer for 1 min to remove zona pellucida followed by 4% PFA fixation. Cells were permeabilized with 1% Triton-X 100 in PBS for 20 min, and blocked with 10% donkey serum. Cells were then incubated with indicated primary antibodies at 4 ℃ overnight followed by staining with Alexa405-, Alexa488-, Alexa594-, and Alexa649-conjugated secondary antibodies. Nuclei were stained with DAPI or PI. Images were obtained with Olympus FV1000 laser-scanning confocal microscopy (Olympus, Japan). The software ImageJ was used for colocalization analysis[61].

**Mass spectrometry**. MBP-CCP6-wt and MBP-CCP6-mut[23,29] proteins were expressed in *E. coli* and purified using the amylose resin (New England BioLabs, Ipswich, USA) according to the manufacturer's instruction. Proteins were immobilized with Affi-gel10 resin to go through OSKM-transduced MEF lysates for affinity chromatography. Eluted fractions were visualized by SDS-PAGE followed by silver staining. Differential bands in SDS-PAGE gels were trypsinized for mass spectrometry with LTQ Orbitrap XL (Thermo Finnigan)[29].

**Immunoprecipitation assay**. MEFs were infected with the indicated retrovirus. Cells were lyzed with ice-cold RIPA buffer (50 mM Tris-HCl [pH 7.4], 150 mM NaCl, 0.5% sodium desoxycholate, 0.1% SDS, 5 mM EDTA, 2 mM PMSF, 20 mg/ml aprotinin, 20 mg/ml leupeptin, 10 mg/ml pepstatin A, 150 mM benzamidine, and 1% Nonidet P-40) for 1 h. Lysates were incubated with the indicated antibodies followed by immunoprecipitation with protein A/G agarose beads and immunoblotting[62]. For in vivo ubiquitination assay, cells were pretreated with MG132 (20 μM) for 4 h and scraped into 100-μl lysis buffer (10 mM Tris-HCl, pH 7.4, 0.5 M NaCl, and 2% SDS). Lysates were heated at 95 ℃ for 10 min and diluted with 10 mM Tris-HCl to reduce SDS concentration to 0.2%. Anti-Klf4 antibody was used to precipitate immunocomplexes, followed by immunoblotting with anti-Ubiquitin antibody. Uncropped scans of results were shown as Supplementary Figure 8.

**In vitro glutamylation assay**. A total of 293 T cells were transfected with expression plasmids of TTLL4, CCP1, and CCP6 for 48 h. Cells were collected and lyzed with PBS–0.2% NP40. Supernatants were incubated with recombinant GST-Klf4 protein at 37 ℃ for 2 h. GST-Klf4 was pulled down with Glutathione Sepharose 4B beads, followed by immunoblotting[29]. Uncropped scans of results were shown as Supplementary Figure 8.

**In vitro ubiquitination reconstitution assay**. Uba1, UbcH5c, Rbx, and ubiquitin were cloned into pET-28a vectors. Klf4 was cloned into pGEX-6p-1 vector. Plasmids were then transformed into *E. coli* strain BL21 (DE3). Protein expression was induced with 1 mM isopropyl-β-D-1-thiogalactopyranoside (IPTG) at 16 ℃ for 24 h. Proteins were purified by Ni-NTA resin column (Novagen) or GST-sepharose column. Cullin1 and Skp1 were cloned into p3×flag-CMV-9 expression vector. βTrCP1 was cloned into pDEST-Flag-PHF13 expression vector. Plasmids were transfected into 293 T cells. Cells were lyzed with RIPA buffer and Flag-tagged proteins were immunoprecipitated with anti-Flag antibody and eluted by Flag-peptide (Sigma-Aldrich). Klf4 was pretreated with TTLL4 prior to in vitro polyglutamylation assay. Before in vitro ubiquitination assay, Klf4 protein was also incubated with recombinant active ERK1 (Millipore) and 2 mM ATP at 30 ℃ for 30 min. In vitro ubiquitination reconstitution assay was performed by mixing E1 (Uba1), E2 (UbcH5c), E3 (βTrCP1, Cullin1, Skp1, and Rbx), ubiquitin, and Klf4 in the ubiquitination buffer (50 mM Tris-HCl, 5 mM $MgCl_2$, 2 mM dithiothreitol, and 2 mM ATP, pH 7.4) at 30 ℃ for 2 h[63]. The uncropped scan of the result was shown as Supplementary Figure 8.

**Gene expression assay by real-time qPCR**. iPSCs and ESCs were sorted by flow cytometry after SSEA-1 staining. A total of $5 \times 10^5$ cells were lyzed in each sample for the analysis below. Total RNAs were extracted with the RNA miniprep kit (LCsciences, Houston, USA) according to the manufacturer's manual. M-MLV reverse transcriptase (Promega, Madison, USA) was used to synthesize cDNA. Quantitative PCR analysis and data collection were performed on the ABI 7300 qPCR system using the primer pairs listed in Supplementary Table 1. Quantitation was normalized to an endogenous β-actin gene[64].

**Microarray analysis**. Mouse ES cell line D3 cells were transfected with the shCCP6 or shCtrl plasmid and cultured in mouse ESC media for 3 days. RNAs from mouse shCCP6 ESCs and shCtrl ESCs were prepared for Affimatrix microarray analysis according to the manufacturer's instruction. Briefly, total RNA was extracted by Trizol reagent. Double-strand cDNA was synthesized by an Invitrogen SuperScript cDNA synthesis kit and labeled according to the Affimatrix protocol. Microarrays were hybridized, washed and scanned according to the manufacturer's instruction. Differentially expressed genes were identified as fold change cutoff >2.0, FDR<0.05.

**DNase I accessibility assay**. SSEA-1$^+$ iPSCs were sorted by flow cytometry. Nuclei were isolated from $1 \times 10^5$ iPSC cells using the nuclei isolating kit (Sigma-Aldrich) according to the manufacturer's protocol. Nuclei were resuspended in 200 μl of DNase I digestion buffer (1 mM EDTA, 0.1 mM EGTA, 5% sucrose, 1 mM $MgCl_2$, and 0.5 mM $CaCl_2$). Equal aliquots of 100 μl of nuclei were treated with the indicated units of DNase I (Sigma, USA) and incubated at 37 ℃ for 5 min. The reactions were stopped by 2 × DNase I stop buffer (20 mM Tris, pH 8.0, 4 mM EDTA, and 2 mM EGTA). DNA was extracted and analyzed by qPCR. The primer pairs aimed at the promoter region for indicated genes are listed below. mNanog promoter: Forward, 5′-ATCCACCTGCCTCTGCCGCCTAA-3′, Reverse, 5′- GCATTGGTGTTTTGCCTGCATGG-3′; mEsrrb promoter: Forward, 5′-CCA ACCAGAAGTGGGTCTTGTTCCT-3′, Reverse, 5′-TGTGGAAGGATCCTGGG ACACAGAT-3′[64].

**Electrophoretic mobility shift assay (EMSA)**. Biotin-labeled double-strand probes were generated by annealing complementary single-stranded oligonucleotides. The probe sequence used for detection was from Nanog promoter region: 5′-GCCGCCTGGGTGCCTGGGAGAATAGGGGGTGGGTAGGGTAGGAGG

CTTG-3′. Recombinant GST-Klf4 and mutants were expressed by *E.coli* and purified as described. LightShift Chemiluminescent EMSA Kit (Thermo Scientific) was used for shift assay according to the manufacturers' instructions. Unlabeled (1-fold or 100-fold of labeled probe) probe was used for competitive reaction[61]. The uncropped scan of the result was shown as Supplementary Figure 8.

**Statistical analysis**. Unless otherwise stated, data are presented as means ± S.D. Student's *t* test was used as statistical analysis by using Microsoft Excel as described.

**Data availability**. The authors declare that all data supporting the findings of this study are available within the article and its supplementary information files or from the corresponding author upon reasonable request. Microarray data that support the findings of this study are available in GEO datasets with the accession code GSE106809.

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

## Acknowledgements

We thank Dr. Xiaohui Wu (University of Fudan) for providing *Ccp6*⁻/⁻ mice. We thank Dr. Degui Chen (Shanghai Institute of Biochemistry and Cell Biology, Chinese Academy of Sciences) for kindly providing βTrCP1 plasmid. We thank Drs. Yang Xu and Tongbiao Zhao (University of California, San Diego) for providing OSKM plasmids. We thank Jing Li (Cnkingbio Company Ltd, Beijing, China) for technical support. We thank Peng Xue, Xiang Ding, Xiang Shi, Yan Teng, Chunli Jiang, Junying Jia, Liangming Yao, Jing Cheng, Yihui Xu, Xudong Zhao, and Xiaofei Guo for technical support. This work was supported by the National Natural Science Foundation of China (31530093, 91640203, 91419308, 31429001, 31470864, 31670886, 81601361, 81572433, 81772646, and 31601189), and the Strategic Priority Research Programs of the Chinese Academy of Sciences (XDB19030203).

## Author contribution

B.Y., B.L., and L.Ha designed and performed the experiments; X.Z. generated *TTLL4*⁻/⁻ mice; B.Y. analyzed data and wrote the paper; S.W., P.X., Y.D., X.Q., Y.W., X.Y., C.L., J.H., P.Z., and L.He analyzed data; L.Y., S.M., G.H., and B.L. performed some experiments; Y.T. initiated the study, crossed mice, and analyzed data; and Z.F. initiated the study, organized, designed, and wrote the paper.

## Additional information

**Competing interests:** The authors declare no competing financial interests.

