## [Peer Review File · Nature Communications]

Reviewers' comments:

Reviewer #1 (Remarks to the Author):

The manuscript by Ye et al. identifies glutamylation of Klf4 as an important step for cellular reprogramming. The author find that increased glutamylation of Klf4 due to deficiency in CCP1 or 6 inhibits K48 ubiquitination of Klf4, thus enhancing its stability and promoting iPSC induction and pluripotency of ESCs. The authors identify the glutamylation site (although it not clear whether this is the only one – please see comments below) and show that its mutation to Ala leads to blastocyst development failure. Furthermore, they identify TTLL1 and 4 as the enzymes responsible for glutamylating Klf4.

Glutamylation is an evolutionarily conserved posttranslational modification that has recently emerged as an important regulator not only of tubulin and the microtubule cytoskeleton (the context in which this modification was initially discovered), but also of other important cellular pathways. There are still only a handful of examples of the physiological mechanism behind glutamylation of non-tubulin substrates and this manuscript contributes one more example in the context of regulating pluripotency. This is an important and interesting contribution and it warrants publication after addressing the comments highlighted below.

1. There seems to be not very precise usage of the term glutamylation and polyglutamyation throughout the manuscript. Polyglutamyation signifies the formation of poly-glutamate chains. The GT335 antibody used in this study detects only one branched glutamate. It thus detects only glutamylation (or monoglutamylation) and NOT polyglutamylation.
2. Related to point 1, the authors should try to use the B3 or ID5 antibodies to determine whether chains longer than one glutamate are added to Klf4. Original reference for this antibody is Rüdiger AH, Rüdiger M, Wehland J, Weber K(1999) Monoclonal antibody ID5: Epitope characterization and minimal requirements for the recognition of polyglutamylated alpha- and beta-tubulin. *Eur J Cell Biol* 78(1): 15–20. Is there any evidence from mass spec that longer chains are generated ? See my comment below about the mass spec data
3. The mass spectra in Figure S3 are not clear. The labels for the peaks are illegible. This figure needs to be improved so that the spectra quality can be evaluated by the reader. Which peaks show glutamylated peptides? The same for ubiquitination. The authors mention that Klf4 has four Glu rich regions. The sequences for these should be presented in SI. Is the region that contains the modified glutamate conserved? Is it homologous to the tubulin tails? Is E381 conserved in other species?
4. The experiment presented in Figure 5D should also be performed for TTLL1 overexpression. Are indeed TTLL1 and TTLL4 modifying the same glutamate on Klf4?
5. The biochemical activities for CCP enzymes is still not understood, however from the overexpression experiments performed so far it seems that CCP1 and 6 remove linear glutamate chains while CCP5 removes the branched glutamate off tubulin tails. These biochemical activities are at odds with what is reported in this ms since the glutamylation of Klf4 is initiated from an internal glutamate ie a branched chain is generated. In order to deglutamylate Klf4, it would seem that a debranching enzyme such as CCP5 is needed. The fact that CCP1 and 6 have an effect suggest that maybe Klf4 is functionalized with longer glutamate chains (see point 2). or maybe the reported biochemical activities for CCP1 and 6 are not as clear cut and they might be able to also remove branched chains. The authors should address these possible discrepancies in their discussion.

6. The CCP1 effect on pluripotency related genes seems to be smaller than for CCP6. There is no mention of the CCP1 litter size. Is it comparable to CCP6-/-? Any thoughts on why CCP6 downregulation has a much stronger effect if the authors propose that they are both involved in removing the same branched glutamate from Klf4?

7. On page 3 – Reference 19 should also be cited as an example of how glutamylation modulates the interaction between microtubules and their partners.

Reviewer #2 (Remarks to the Author):

The manuscript by Ye et al. identifies Klf4 polyglutamylation as a requirement for iPSC reprogramming and early embryonic development. Stabilization of Klf4 by polyglutamylation has not been reported to date and thus represents a novel finding. Molecular mechanisms of action (impact on Klf4 stability by antagonizing ubiquitination) are provided and the different genetic and small molecule manipulations are consistent. The findings of this manuscript highlight the importance of posttranslational modifications (PTM) in reprogramming events – and will thus be of insight to readers interested in pluripotency, as well as PTM research.

The evidence provided at the molecular and cellular level are convincing. While these findings are exciting, the manuscript sadly does not analyze the embryonic phenotypes of the numerous mutants generated in sufficient detail. If addressed sufficiently, this manuscript will provide a better picture of the overall function of Klf4 polyglutamylation. Mentioning rather briefly embryonic phenotypes without characterizing them makes the manuscript not acceptable for publication in its present state. In addition, the manuscript suffers from a lack of integration with previous data on Klf4 function in embryos and ESCs, as well as several vague points in the text and methods that need to be clarified.

Major Comments:

1. In order to support the claim of Klf4 glutamylation being required for normal development in vivo, a number of experiments are required for the Klf4-E381R mutants (and relevant controls):
 - a. Quantification of embryos/genotypes at several stages of development (e.g. preimplantation days, E7.5, E15.5, E18.5) from heterozygous intercrosses to establish the time-point of lethality.
 - b. Detailed phenotype analysis: Staining of preimplantation embryos for cell fate markers (e.g. Cdx2, Nanog, Gata) as well as Klf4, and analysis of counts for each cell fate. Staining for apoptosis markers would also be insightful.
2. The analysis above is particularly important given that the Klf4 knockout mice die shortly after birth (Segre Nature Genetics 1999). Why then would the Klf4-E381R point mutant be preimplantation lethal, as claimed here? It is hard to imagine it can act as a dominant negative, given that the heterozygous mice for the point mutation are viable. Could this be due to dimerization between Klf4-E381R and other Klf4s? The authors would need to investigate that. For example, it is known that Klf2 and 5 are redundant with Klf4 in ES cells (Jiang NCB 2008) and iPSCs (Nakagawa Nat Biotech 2008). These findings from other labs need to be both discussed and addressed experimentally.
3. Similar to point 1), the phenotypes of the CCP1/6 and TTL1/4 preimplantation embryos require a more detailed analysis: Staining of preimplantation embryos for cell fate markers could be sufficient here.

4. Page 6: Intra-oviductal injections of small molecules seem inappropriate to answer this question. The inhibitors could have effects on uterine tissues that cause the phenotype. Further, the method is not well described in the methods. According to the reference given, the injections would have been performed prior to fertilization, which could have effects on the gametes or the zygote that are not relevant for the proposed function in development of the ICM. This experiment seems confusing and unnecessary.

5. The manuscript is difficult to read and at some points misleading due to language issues (sentence construction, etc.). Thus, the manuscript could greatly benefit from being edited by an English native speaker.

Minor Comments (Text):

1. Page 3: 'O-GlcNacylation (...) cell reprogramming.' Please also cite PMID: 26949256 (Elife. 2016 Mar 7;5:e10647)

2. What are the growth rates of CCP1^{-/-} and CCP6^{-/-} MEFs? Changes in MEF growth rates could affect reprogramming efficiencies.

3. Page 5: References 35 and 36 are not good references for the statement of the sentence.

4. Page 6 (and onward): As it is not mentioned in the methods: When the authors refer to E3.0, they analyzed embryos in the middle of the night? Please clarify.

5. Page 6: 'zygote reprogramming and pluripotency formation': here and below: do not use the term "reprogramming" on the early embryo, as it could be confused with SCNT. Just say early development and specify stage.

6. Page 8: Regarding colocalization of GT335 and Klf4: There seems to be some granularity to the staining (i.e. not the entire nucleus is stained). Could you assess sub-nuclear co-localization at higher magnification?

7. Page 8: please re-phrase to 'co-localization' instead of Klf4 polyglutamylation, since this is not established

8. Page 9: 'resulted in its degradation in 4-8 cell stage embryos': Please mention the phenotype of the embryos or at least "see below"

9. Figure 5B: 232R mutation does not seem to increase colocalization number

10. Page 11: 'augmented ESC pluripotency' – this vague term should be avoided. Increased Self-renewal may be a better term (if it is statistically significant)

11. Page 12: 'suppressed blastocyst development' – this term is also too vague. Please describe in more detail (see major point 1)

12. General point: The discussion is too long, and repetitive with the introduction. Please shorten and put your new findings into context. In particular: Why is the Klf4-E381R mutant blastocyst lethal when the KO is perinatal lethal? And could there be other targets for polyglutamylation?

13. Methods: Reference or give more detail on Ccp6 mutant generation
14. Methods page 17: has KSOM been used as a flushing media? KSOM is standard for embryo culture, but should not be used for flushing of embryos (better: M2 or similar)
15. Methods page 17: Flushed at E3.0 – why not at E3.5? was this during the night or the day?
16. Methods page 17: Embryos were fixed, sectioned for staining with H&E? There is no such data in the figures.
17. Methods page 17: please share guide sequences for the generation of Ttll1 and Ttll4 ko mice. Please provide guide and template sequences for Klf4 point mutation.

Minor Comments (Figures):

18. Figure 1 & 6D: The image quality is not good enough to highlight embryo phenotypes. For quantification, please indicate the number of embryos as well.
19. Figure 1G: labeling is confusing: was intro-oviductal injection at E3.0? and flush at E3.5?
20. Figures 1D, 2F: how many teratomas were analyzed per condition, and are they all from the same cell line? Ideally they would be at least 2 cell lines per genotype, and 3-4 teratomas per line per timepoint.
21. Figures 2A-B, 5B: please use statistics to address potential significance
22. Figure 2C, 3F-G, S3L: please add number of embryos/cells analyzed
23. Figure 5F-H: change allele description from mut to E381R, otherwise misleading as KO
24. Figure S1A: Is this a maternal, paternal or zygotic effect? Do you see the same effect if you cross null females with wt males (and reciprocal)?
25. Figure S1D-E: Pluripotency genes should not even be detectable in MEFs
26. Figure S3K: please add quantification

Reviewer #3 (Remarks to the Author):

Ye et al. provide evidence that protein polyglutamylation is implicated in cell reprogramming and early embryo development. Depletion of the polyglutamylases TTLL4 and TTLL1 inhibits iPS colony formation from mouse and human somatic cells and murine blastocyst formation. The same phenotype is mimicked by overexpression of two carboxypeptidases, CCP1 and CCP6.

The authors go on to show that the main target of polyglutamylation during cell reprogramming is the pluripotency transcription factor Klf4. Polyglutamylation of Klf4 at Glu381 protects the protein from ubiquitination and rapid degradation, favoring its accumulation in reprogramming cells and, in turns, the positive regulation of other pluripotency genes such as Nanog and Esrrb.

This report provides a novel mechanism that regulates the balance of pluripotency transcription

factors and will certainly be of interest to readers in the field of stem cell and reprogramming. However, few questions arise from the manuscript that need to be address before this study can be considered for publication.

Major criticisms

1. All iPS reprogramming experiment were performed using individual retroviral vector to overexpress the 4 Yamanaka factors and efficiency was scored based on AP staining. It is well known that the use of retroviral vectors causes the formation of partially reprogrammed clones which positively stained for AP. Considering the small differences in efficiency throughout the paper (3-4 fold), the authors should reproduce the reprogramming experiments using a dox inducible system (secondary system or inducible lentiviral vectors) and score for efficiency after dox removal with Nanog staining to select for bona fide iPSCs.
2. One of the most interesting observation is the higher expression of pluripotency genes in ES/iPS cells KD for Ccp1 or Ccp6. Considering the low expression of these two gene in ESCs ("almost undetectable"), how do the authors explain such a remarkable effect? It would be important to reproduce the result in a different ESC line and to perform global gene expression in ESC depleted of Ccp1 and Ccp6 to verify the specific effect of these depletions on the pluripotency transcriptional network.
3. Considering the higher expression of pluripotency markers in ES depleted of Ccp1 and Ccp6 I would have expected a delay or impairment in ES differentiation. Is this the case? Moreover, the authors show that teratomas are bigger. Is this due to the presence of undifferentiated ES cells?
4. The authors should assess the effect of Ccp1 and Ccp6 depletion and overexpression on MEF and ESC viability and proliferation.
5. The GT335 accumulation in almost 100% of the cell nuclei during reprogramming shown in Fig 3A is surprising considering that the authors are performing all the experiments by infecting MEFs with individual retroviruses. The total % of infected MEFs with all the four factors at the same time is most likely lower than 100%. How do the authors explain the accumulation of GT335 in cells that are not undergoing reprogramming?
6. Considering the role of Klf4 during the transition from primed to naïve pluripotency, how does Ccp1 and Ccp6 depletion affect pluripotency in ESC cultured in naïve conditions?
7. It would be important to show how Ccp1, Ccp6, TTLL1 and TTLL4 expression changes during reprogramming and if these genes are directly regulated by the OSKM or SNEL factors by looking at published RNAseq and CHIPseq datasets.
8. The higher DNase sensitivity at the Nanog promoter shown in figure 5C is probably due to the higher amount of iPS cells present after 3 weeks of reprogramming in Ccp6 KO cells respect to the other samples. The authors should perform again the DNase assay for the different conditions in a pure population of ES/iPS cells.
9. To definitively prove that the change in Klf4 stability is directly causing the increase in pluripotency gene expression observed in ccp1 and ccp6 KO ES cells, the authors should perform Klf4 ChIP-qPCR at regulatory regions of the genes shown in Supp Fig1B.
10. Considering the role of Ccp1 and Ccp6 as roadblock for iPS reprogramming, it would be interesting to check their expression dynamics (together with TTLL1 And TTLL4) during deterministic iPS

reprogramming (datasets from Rais et al Nature 2013 and Di Stefano et al Nature 2014)

Minor criticisms

1. In Figure 1C the authors should use the appropriate isotype control instead of the unstained sample in gray.
2. The gene Sall4 is not always spelled correctly.
3. The shKlf4 efficiency in ESCs is not shown in the text.
4. The authors miss to reference an important study on the role of Klf4 and protein stability in iPS reprogramming (Di Stefano et al., Nature Cell Biology 2016).

Point-by-point response to the reviewers' comments

Reviewer #1

The manuscript by Ye et al. identifies glutamylation of Klf4 as an important step for cellular reprogramming. The author find that increased glutamylation of Klf4 due to deficiency in CCP1 or 6 inhibits K48 ubiquitination of Klf4, thus enhancing its stability and promoting iPSC induction and pluripotency of ESCs. The authors identify the glutamylation site (although it not clear whether this is the only one – please see comments below) and show that its mutation to Ala leads to blastocyst development failure. Furthermore, they identify TTL1 and 4 as the enzymes responsible for glutamylating Klf4. Glutamylation is an evolutionarily conserved posttranslational modification that has recently emerged as an important regulator not only of tubulin and the microtubule cytoskeleton (the context in which this modification was initially discovered), but also of other important cellular pathways. There are still only a handful of examples of the physiological mechanism behind glutamylation of non-tubulin substrates and this manuscript contributes one more example in the context of regulating pluripotency. This is an important and interesting contribution and it warrants publication after addressing the comments highlighted below.

1. There seems to be not very precise usage of the term glutamylation and polyglutamylation throughout the manuscript. Polyglutamylation signifies the formation of poly-glutamate chains. The GT335 antibody used in this study detects only one branched glutamate. It thus detects only glutamylation (or monoglutamylation) and NOT polyglutamylation.

Answer: We also used B3 antibody to detect Klf4 glutamylation from iPSC lysates and found Klf4 glutamylation was also detectable (new Fig. S4B), indicating that Klf4 glutamylation was polyglutamylation. We addressed this point in the revised text.

2. Related to point 1, the authors should try to use the B3 or ID5 antibodies to determine whether chains longer than one glutamate are added to Klf4. Original reference for this antibody is Rüdiger AH, Rüdiger M, Wehland J, Weber K(1999) Monoclonal antibody ID5: Epitope characterization and minimal requirements for the recognition of polyglutamylated alpha- and beta-tubulin. Eur J Cell Biol 78(1):15–20. Is there any evidence from mass spec that longer chains are generated? See my comment below about the mass spec data

Answer: This is a good suggestion. As we addressed above, B3 antibody could detect the glutamate chains of Klf4. As shown in the Fig. 3C, we used enzymatically inactive mutant of CCP6 (CCP6mut) to pull down its candidate substrates, followed by mass spectrometry. The glutamate chains of Klf4 were not detected by our mass spec assays. The possible reasons could be: 1) Klf4 was easily degraded; 2) there were not enough amounts of glutamate peptides for mass spec; 3) for mass spectrometry, proteins were cleaved by trypsin to generate short peptides. The picked peptides might not contain modified ones; and 4) our mass spec could have technical problems to detect modifications.

3. The mass spectra in Figure S3 are not clear. The labels for the peaks are illegible. This

figure needs to be improved so that the spectra quality can be evaluated by the reader. Which peaks show glutamylated peptides? The same for ubiquitination. The authors mention that Klf4 has four Glu rich regions. The sequences for these should be presented in SI. Is the region that contains the modified glutamate conserved? Is it homologous to the tubulin tails? Is E381 conserved in other species?

Answer: We provided better mass spectra in the new Fig. S3E. As we addressed the reasons in the Question#2, the picked peptides in mass spectra did not include glutamylated or ubiquitinated peptides. We provided this sequence information in the new Fig. S4C. Yes, the region containing E381 was highly conserved from zebrafish to human (Fig. S4C). However, this region was not homologous to the tubulin tails.

4. The experiment presented in Figure 5D should also be performed for TLL1 overexpression. Are indeed TLL1 and TLL4 modifying the same glutamate on Klf4?

Answer: We conducted these experiments by overexpressing Flag-tagged TLL1 together with OSKM factors in MEFs.. Similar results were achieved (Fig. S4D and Fig. S5O). We conclude that TLL1 and TLL4 can modify the same glutamate on Klf4. We addressed this point in the revised text.

5. The biochemical activities for CCP enzymes is still not understood, however from the overexpression experiments performed so far it seems that CCP1 and 6 remove linear glutamate chains while CCP5 removes the branched glutamate off tubulin tails. These biochemical activities are at odds with what is reported in this ms since the glutamylation of Klf4 is initiated from an internal glutamate ie a branched chain is generated. In order to deglutamylate Klf4, it would seem that a debranching enzyme such as CCP5 is needed. The fact that CCP1 and 6 have an effect suggest that maybe Klf4 is functionalized with longer glutamate chains (see point 2). or maybe the reported biochemical activities for CCP1 and 6 are not as clear cut and they might be able to also remove branched chains. The authors should address these possible discrepancies in their discussion.

Answer: That is a very good point. We showed that Klf4 virtually underwent polyglutamylation in iPSCs. CCP1 and CCP6 could synergistically shorten longer glutamate chains of glutamylated Klf4, while CCP5 may be needed to deglutamylate Klf4 at a branching glutamate site. Alternatively, CCP1 and CCP6 could also remove a branched glutamate of glutamylated Klf4, whose biochemical activities could be different from tubulins (Rogowski K et al. Cell, 2010). We addressed these possible discrepancies in the discussion section of our revised manuscript.

6. The CCP1 effect on pluripotency related genes seems to be smaller than for CCP6. There is no mention of the CCP1 litter size. Is it comparable to CCP6-/-? Any thoughts on why CCP6 downregulation has a much stronger effect if the authors propose that they are both involved in removing the same branched glutamate from Klf4?

Answer: We repeated these experiments with another primer sets for Ccp1 and obtained similar results compared to CCP6 (new Fig. S1B, S1D and S2H). The previous discrepancies could be caused by non-efficient primers for qPCR assay. Since homozygous *Ccp1*-deficient mice are male sterile, we could not compare the litter size

with littermate control mice. However, CCP1 deficiency was really comparable to CCP6 deficiency for iPSC formation, teratomas formation as well as ES clone pluripotency.

7. On page 3 – Reference 19 should also be cited as an example of how glutamylation modulates the interaction between microtubules and their partners.

Answer: We addressed this reference as suggested.

Reviewer #2

The manuscript by Ye et al. identifies Klf4 polyglutamylation as a requirement for iPSC reprogramming and early embryonic development. Stabilization of Klf4 by polyglutamylation has not been reported to date and thus represents a novel finding. Molecular mechanisms of action (impact on Klf4 stability by antagonizing ubiquitination) are provided and the different genetic and small molecule manipulations are consistent. The findings of this manuscript highlight the importance of posttranslational modifications (PTM) in reprogramming events – and will thus be of insight to readers interested in pluripotency, as well as PTM research. The evidence provided at the molecular and cellular level are convincing. While these findings are exciting, the manuscript sadly does not analyze the embryonic phenotypes of the numerous mutants generated in sufficient detail. If addressed sufficiently, this manuscript will provide a better picture of the overall function of Klf4 polyglutamylation. Mentioning rather briefly embryonic phenotypes without characterizing them makes the manuscript not acceptable for publication in its present state. In addition, the manuscript suffers from a lack of integration with previous data on Klf4 function in embryos and ESCs, as well as several vague points in the text and methods that need to be clarified.

Major Comments:

1. In order to support the claim of Klf4 glutamylation being required for normal development in vivo, a number of experiments are required for the Klf4-E381R mutants (and relevant controls):

a. Quantification of embryos/genotypes at several stages of development (e.g. preimplantation days, E7.5, E15.5, E18.5) from heterozygous intercrosses to establish the time-point of lethality.

Answer: This is a good suggestion. Klf4-E381A knockin mice displayed defective blastocyst development and caused embryonic lethality at preimplantation E3.5 stage. We provided these data in the new Fig. 5G.

b. Detailed phenotype analysis: Staining of preimplantation embryos for cell fate markers (e.g. Cdx2, Nanog, Gata) as well as Klf4, and analysis of counts for each cell fate. Staining for apoptosis markers would also be insightful.

Answer: We provided these data in the new Fig. 5H and 5I.

2. The analysis above is particularly important given that the Klf4 knockout mice die shortly after birth (Segre Nature Genetics 1999). Why then would the Klf4-E381R point mutant be preimplantation lethal, as claimed here? It is hard to imagine it can act as a dominant negative, given that the heterozygous mice for the point mutation are viable.

Could this be due to dimerization between Klf4-E381R and other Klf5s? The authors would need to investigate that. For example, it is known that Klf2 and 5 are redundant with Klf4 in ES cells (Jiang NCB 2008) and iPSCs (Nakagawa Nat Biotech 2008). These findings from other labs need to be both discussed and addressed experimentally.

Answer: This is a very good point. We found that Klf4-E381A was really dimerized with Klf5 in ESCs, while Klf4-wt protein did not (Attached Fig. 1). In Klf4-E381A KI mice, the dimerization between Klf4-E381A and Klf5 could inactivate the Klf5 function as well, leading to preimplantation lethality. How Klf4-E381A point mutation affects the function of Klf5 still needs to be further investigated

3. Similar to point 1), the phenotypes of the CCP1/6 and TTL1/4 preimplantation embryos require a more detailed analysis: Staining of preimplantation embryos for cell fate markers could be sufficient here.

Answer: We provided these data in the new Fig. S8C-S8F.

4. Page 6: Intra-oviductal injections of small molecules seem inappropriate to answer this question. The inhibitors could have effects on uterine tissues that cause the phenotype. Further, the method is not well described in the methods. According to the reference given, the injections would have been performed prior to fertilization, which could have effects on the gametes or the zygote that are not relevant for the proposed function in development of the ICM. This experiment seems confusing and unnecessary.

Answer: We deleted these data in our revised manuscript.

5. The manuscript is difficult to read and at some points misleading due to language issues (sentence construction, etc.). Thus, the manuscript could greatly benefit from being edited by an English native speaker.

Answer: We carefully revised our whole manuscript and corrected grammatical errors throughout the text.

Minor Comments (Text):

1. Page 3: 'O-GlcNacylation (...) cell reprogramming.' Please also cite PMID: 26949256 (Elife. 2016 Mar 7;5:e10647)

Answer: We cited this reference accordingly.

2. What are the growth rates of CCP1^{-/-} and CCP6^{-/-} MEFs? Changes in MEF growth rates could affect reprogramming efficiencies.

Answer: We tested growth rates of indicated MEFs via CCK-8 staining. We observed that the growth rates of CCP1^{-/-} and CCP6^{-/-} MEFs were comparable to their WT counterpart MEFs (Attached Fig. 2A)

3. Page 5: References 35 and 36 are not good references for the statement of the sentence.

Answer: We changed these references with appropriate ones.

4. Page 6 (and onward): As it is not mentioned in the methods: When the authors refer to E3.0, they analyzed embryos in the middle of the night? Please clarify.

Answer: We made wrong descriptions. E3.5 embryos were flushed from uteri about 96 hours post hCG injection. We changed E3.0 to E3.5 throughout the text.

5. Page 6: 'zygote reprogramming and pluripotency formation': here and below: do not use the term "reprogramming" on the early embryo, as it could be confused with SCNT. Just say early development and specify stage.

Answer: We revised this wording in the revised text.

6. Page 8: Regarding colocalization of GT335 and Klf4: There seems to be some granularity to the staining (i.e. not the entire nucleus is stained). Could you assess sub-nuclear co-localization at higher magnification?

Answer: We replaced the old images with better ones (new Fig. 3F).

7. Page 8: please re-phrase to 'co-localization' instead of Klf4 polyglutamylation, since this is not established

Answer: We revised this wording in the revised text.

8. Page 9: 'resulted in its degradation in 4-8 cell stage embryos': Please mention the phenotype of the embryos or at least "see below"

Answer: We revised this wording in the revised text.

9. Figure 5B: 232R mutation does not seem to increase colocalization number

Answer: We repeated AP staining and included new data in the new Fig. 5B.

10. Page 11: 'augmented ESC pluripotency' – this vague term should be avoided. Increased Self-renewal may be a better term (if it is statistically significant)

Answer: We revised this wording in the revised text.

11. Page 12: 'suppressed blastocyst development' – this term is also too vague. Please describe in more detail (see major point 1)

Answer: We stained preimplantation embryos for cell fate markers (Nanog for epiblast (EPI), Gata6 for primary endoderm (PrE), Cdx2 for trophectoderm (TE)) as well as apoptosis marker cleaved- caspase 3 (cl-Casp 3), and analysis of counts for each cell fate. TTLL1 and TTLL4 deficiency suppressed blastocyst development, leading to decreased blastocoel area, disordered lineage marker expression and increased apoptotic cells in the indicated preimplantation embryos. We provided these data in the new Fig. S8C.

12. General point: The discussion is too long, and repetitive with the introduction. Please shorten and put your new findings into context. In particular: Why is the Klf4-E381R mutant blastocyst lethal when the KO is perinatal lethal? And could there be other targets for polyglutamylation?

Answer: As we addressed in the Major Question#2, In Klf4-E381A KI mice, the dimerization between Klf4-E381A and Klf5 could inactivate the Klf5 function, leading to preimplantation lethality. Yes, there could be other targets for glutamylation in early embryo development. We revised the discussion section and addressed the above point as well.

13. Methods: Reference or give more detail on Ccp6 mutant generation

Answer: We provided the reference on CCP6 mutant generation in the revised text.

14. Methods page 17: has KSOM been used as a flushing media? KSOM is standard for embryo culture, but should not be used for flushing of embryos (better: M2 or similar)

Answer: We are sorry that we made wrong descriptions. In fact, we used M2 medium for embryo flushing. We changed it accordingly.

15. Methods page 17: Flushed at E3.0 – why not at E3.5? was this during the night or the day?

Answer: Flushed at E3.5, but not E3.0. We corrected it in the revised text.

16. Methods page 17: Embryos were fixed, sectioned for staining with H&E? There is no such data in the figures.

Answer: We deleted this sentence.

17. Methods page 17: please share guide sequences for the generation of Ttll1 and Ttll4 ko mice. Please provide guide and template sequences for Klf4 point mutation.

Answer: We provided all the messages in the methods section.

Minor Comments (Figures):

18. Figure 1 & 6D: The image quality is not good enough to highlight embryo phenotypes. For quantification, please indicate the number of embryos as well.

Answer: We repeated these experiments and provided better images. We also provided the number of embryos.

19. Figure 1G: labeling is confusing: was intro-oviductal injection at E3.0? and flush at E3.5?

Answer: We deleted these data as suggested.

20. Figures 1D, 2F: how many teratomas were analyzed per condition, and are they all from the same cell line? Ideally they would be at least 2 cell lines per genotype, and 3-4 teratomas per line per timepoint.

Answer: 6 teratomas were analyzed pre condition. These teratomas were from 2 iPSC cell lines. We addressed this point in the new figure legend.

21. Figures 2A-B, 5B: please use statistics to address potential significance

- Answer:** We performed statistics for these results in the new Fig. 2A-B and 5B.
22. Figure 2C, 3F-G, S3L: please add number of embryos/cells analyzed
Answer: We added the numbers of embryos in the new Fig. 2C, 3F-G and S4J.
23. Figure 5F-H: change allele description from mut to E381R, otherwise misleading as KO
Answer: We changed this description.
24. Figure S1A: Is this a maternal, paternal or zygotic effect? Do you see the same effect if you cross null females with wt males (and reciprocal)?
Answer: It could be a zygotic effect. Higher litter size at birth was only observed when null females and null males were crossed.
25. Figure S1D-E: Pluripotency genes should not even be detectable in MEFs
Answer: This is the case. In Fig. S1D, expression levels of pluripotency genes were detected in iPSCs generated from indicated WT and CCP1/6 KO MEFs. We deleted the Fig. S1E.
26. Figure S3K: please add quantification
Answer: We added quantification in the new Fig. S4I.

Reviewer #3

Ye et al. provide evidence that protein polyglutamylation is implicated in cell reprogramming and early embryo development. Depletion of the polyglutamylases TTLL4 and TTLL1 inhibits iPS colony formation from mouse and human somatic cells and murine blastocyst formation. The same phenotype is mimicked by overexpression of two carboxypeptidases, CCP1 and CCP6. The authors go on to show that the main target of polyglutamylation during cell reprogramming is the pluripotency transcription factor Klf4. Polyglutamylation of Klf4 at Glu381 protects the protein from ubiquitination and rapid degradation, favoring its accumulation in reprogramming cells and, in turns, the positive regulation of other pluripotency genes such as Nanog and Esrrb. This report provides a novel mechanism that regulates the balance of pluripotency transcription factors and will certainly be of interest to readers in the field of stem cell and reprogramming. However, few questions arise from the manuscript that need to be address before this study can be considered for publication.

Major criticisms:

1. All iPS reprogramming experiment were performed using individual retroviral vector to overexpress the 4 Yamanaka factors and efficiency was scored based on AP staining. It is well known that the use of retroviral vectors causes the formation of partially reprogrammed clones which positively stained for AP. Considering the small differences in efficiency throughout the paper (3-4 fold), the authors should reproduce the reprogramming experiments using a dox inducible system (secondary system or inducible lentiviral vectors) and score for efficiency after dox removal with Nanog staining to select for bona fide iPSCs.

Answer: This is a good suggestion. We isolated MEFs from E14.5 embryos of *4F2A* mice for reprogramming experiments. With 3 week induction, we performed Nanog staining after doxycycline removal. We provided these data in the new Fig. 1E, 5A and 6A.

2. One of the most interesting observation is the higher expression of pluripotency genes in ES/iPS cells KD for *Ccp1* or *Ccp6*. Considering the low expression of these two gene in ESCs (“almost undetectable”), how do the authors explain such a remarkable effect? It would be important to reproduce the result in a different ESC line and to perform global gene expression in ESC depleted of *Ccp1* and *Ccp6* to verify the specific effect of these depletions on the pluripotency transcriptional network.

Answer: We also repeated these experiments with a different ES R1 cell line and obtained similar results (data not shown). We performed gene expression profile chip assay in WT and CCP6 depleted ESCs. We noticed that CCP6 knockdown in ESCs virtually caused elevated expression of pluripotency transcriptional network (new Fig. S2E).

3. Considering the higher expression of pluripotency markers in ES depleted of *Ccp1* and *Ccp6* I would have expected a delay or impairment in ES differentiation. Is this the case? Moreover, the authors show that teratomas are bigger. Is this due to the presence of undifferentiated ES cells?

Answer: This is not the case. Knockdown of CCP1 or CCP6 could normally generated 3 germ layer genes upon RA-induced ectoderm differentiation, bFGF/sodium butyrate-induced endoderm differentiation as well as OP9/Flt3L-induced mesoderm differentiation (Attached Fig. 3A). In addition, pups of *Ccp6*^{-/-} mice developed normally. We further dissociated teratomas into single cell suspension, stained with anti-Nanog antibody for FACS analysis. Nanog-positive ES cells were undetectable in CCP1 and CCP6 depleted ES cells (Attached Fig. 3B). These data indicate that CCP1 or CCP6 depletion does not affect ESC differentiation.

4. The authors should assess the effect of *Ccp1* and *Ccp6* depletion and overexpression on MEF and ESC viability and proliferation.

Answer: We found that depletion and overexpression of CCP1 and CCP6 in MEFs did not affect apoptosis (cl-casp3, Attached Fig. 2B) and growth rates of MEFs (Attached Fig. 2C). For ESCs, depletion of CCP1 or CCP6 promoted cell proliferation, whereas overexpression of CCP1 or CCP6 suppressed cell proliferation (Attached Fig. 2D)

5. The GT335 accumulation in almost 100% of the cell nuclei during reprogramming shown in Fig 3A is surprising considering that the authors are performing all the experiments by infecting MEFs with individual retroviruses. The total % of infected MEFs with all the four factors at the same time is most likely lower than 100%. How do the authors explain the accumulation of GT335 in cells that are not undergoing reprogramming?

Answer: We used a single lentiviral reprogramming vector termed STEMCCA for iPSC induction. We virtually obtained near 100% infection efficiency. We provided this information in the methods section of our revised manuscript. During the early stage of reprogramming induction, accumulation of GT335 really appeared in the nucleus of MEFs that were not undergoing reprogramming. We assume that other substrates may undergo glutamylation for iPSC induction besides Klf4. We are still identifying the other glutamylation substrates during cell reprogramming.

6. Considering the role of Klf4 during the transition from primed to naïve pluripotency, how does Ccp1 and Ccp6 depletion affect pluripotency in ESC cultured in naïve conditions?

Answer: Mouse ESCs were grown in the presence of the differentiation stimuli inhibitors (Nichols J and Smith. A, Cell Stem Cell, 2009). We cultured ESCs using 2i plus LIF conditions. We found that depletion of CCP1 or CCP6 enhanced self-renewal of ESCs in naïve conditions (Attached Fig.4).

7. It would be important to show how Ccp1, Ccp6, TTLL1 and TTLL4 expression changes during reprogramming and if these genes are directly regulated by the OSKM or SNEI factors by looking at published RNAseq and ChIPseq datasets.

Answer: This is a good suggestion. We analyzed RNAseq dataset GSE45352 (Rais.Y. et al. Nature, 2013) for OSKM-induced reprogramming. We found that Ccp1 (encoded by *Agtpbp1* gene) was downregulated and *Ttll4* was upregulated over dox induced OSKM expression (Fig. S2F). We also analyzed ChIPseq datasets (GSE90893: GSM2417130--GSM2417133 for 48 hr timepoint of reprogramming; GSM2417134--GSM2417137 for pre-iPSC formation) (Chronis C. et al. Cell, 2017). We noticed that Klf4 accumulated to *Ttll1* promoter upon pre-iPSC formation. Moreover, Oct4, Sox2, Klf4 and c-Myc accumulated to *Ttll4* promoter in OSKM-expressed MEFs and in pre-iPSCs. Whereas Ccp6 was not directly regulated by OSKM from these datasets. cMyc and Klf4 accumulated to CCP1 promoter in pre-iPSCs. We provided these data in the Attached Fig. 5. In addition, since CCP1/6 and TTLL1/4 exert their functions by enzymatic activities against their respective substrates, their enzymatic activities are also modulated by posttranslational modifications and other regulatory factors. We addressed this point in the discussion sections of our revised manuscript.

8. The higher DNase sensitivity at the Nanog promoter shown in figure 5C is probably due to the higher amount of iPS cells present after 3 weeks of reprogramming in Ccp6 KO cells respect to the other samples. The authors should perform again the DNase assay for the different conditions in a pure population of ES/iPS cells.

Answer: We sorted SSEA-1 positive ES cells and performed DNase I accessibility assay using equal numbers of cells. We provided these data in the new Fig. 5C.

9. To definitively prove that the change in Klf4 stability is directly causing the increase in pluripotency gene expression observed in *ccp1* and *ccp6* KO ES cells, the authors should perform Klf4 ChIP-qPCR at regulatory regions of the genes shown in Supp Fig1B.

Answer: We performed Klf4 ChIP-qPCR at regulatory regions of the indicated genes and provided these data in the new Fig. S5F.

10. Considering the role of Ccp1 and Ccp6 as roadblock for iPS reprogramming, it would be interesting to check their expression dynamics (together with TTLL1 And TTLL4) during deterministic iPS reprogramming (datasets from Rais et al Nature 2013 and Di Stefano et al Nature 2014)

Answer: We analyzed Ccp1 and Ccp6 expression dynamics during iPSC reprogramming. Based on analysis of RNAseq dataset GSE45352 (Rais J et al, Nature, 2013), we found that Ccp1 (also called Agtpbp1) was downregulated and Ttll4 was upregulated over dox induced OSKM expression (Fig. S2F). From RNAseq dataset GSE52396 (Di Stefano B, et al., Nature, 2014), Ccp6 (also called Agbl4) was downregulated during early reprogramming induction (Fig. S2F). In addition, since CCP1/6 and TTLL1/4 exert their functions by enzymatic activities against their respective substrates, their enzymatic activities are also modulated by posttranslational modifications and other regulatory factors. We addressed this point in the discussion sections of our revised manuscript.

Minor criticisms:

1. In Figure 1C the authors should use the appropriate isotype control instead of the unstained sample in gray.

Answer: We actually used isotype control of mouse IgM, but we made wrong labeling. We corrected it in the new Fig. 1C.

2. The gene Sall4 is not always spelled correctly.

Answer: We corrected it.

3. The shKlf4 efficiency in ESCs is not shown in the text.

Answer: We provided these data in the new Fig. S5D.

4. The authors miss to reference an important study on the role of Klf4 and protein stability in iPS reprogramming (Di Stefano et al., Nature Cell Biology 2016).

Answer: We cited this reference in our revised text.

Attached Figure 1. Klf4-E381A mutant is dimerized with Klf5 in ESCs. ESCs were lysed by RIPA lysis buffer and incubated with purified GST-Klf4-wt or GST-Klf4-E381A mutant protein, followed by GST beads pull-down assay. Data are representative of three independent experiments.

Attached Figure 2. CCP1 and CCP6 affect ESC proliferation, but not MEFs. (A) CCP1 or CCP6 deficiency did not affect MEF proliferation via CCK-8 staining. (B) CCP1/CCP6 depletion or overexpression did not affect apoptosis of MEFs. Cells were stained with cleaved caspase3 (cl-Casp3) and analyzed by flow cytometry. (C) CCP1/CCP6 depletion or overexpression did not affect MEF proliferation via CCK-8 staining. (D) CCP1/CCP6 depletion promoted proliferation of ESCs. Overexpression of CCP1/CCP6 inhibited proliferation of ESCs. Data are representative of three independent experiments.

Attached Figure 3. Depletion of CCP1 or CCP6 in ESCs does not delay differentiation. (A). ESCs depleted of Ccp1 or Ccp6 can normally express 3 germ layer genes upon RA-induced ectoderm differentiation, bFGF/sodium butyrate-induced endoderm differentiation and OP9/Flt3L-induced mesoderm differentiation. (B) Teratomas were dissociated into single cell suspension, stained with anti-Nanog antibody and analyzed by flow cytometry. Data are representative of three independent experiments.

Attached Figure 4. Depletion of CCP1 or CCP6 enhances self-renewal of ESCs in a naïve condition. Mouse ESCs were cultured in 2i plus LIF conditions for 5 days, followed by AP staining. Data are representative of three independent experiments.

Attached Figure 5. TLL4 and CCP1 are directly regulated by OSKM during reprogramming. ChIPseq datasets (GSE90893) showed that Klf4 was accumulated to Tll1 promoter in pre-iPSCs. Oct4, Sox2, Klf4 and c-Myc were accumulated to Tll4 promoter in OSKM-expressed MEFs and in pre-iPSCs. Ccp6 was not directly regulated by OSKM. cMyc and Klf4 were accumulated to Ccp1 promoter in pre-iPSCs.

REVIEWERS' COMMENTS:

Reviewer #1 (Remarks to the Author):

The authors have addressed my concerns.

Reviewer #2 (Remarks to the Author):

The authors have address most of the concerns we raised. The points below should still be addressed:

1. Regarding the phenotype of the Klf4 point mutant:

- a. 5G; labelling is misleading; "alive at E3.5" is not necessarily normal development - stage(s) that the embryos reached at 3.5 should be reported;
- b. "Primitive endoderm" is the correct designation, not "primary endoderm";
- c. 5H, I: the quantification is convincing, but the images shown are of poor quality; can better images be show?

2. On the important issue that the Klf4 point-mutant can pull down Klf5, suggestive of a potential reason for the more severe phenotype than Klf4^{-/-} mice: these data need to be discussed and show in a supplementary figure, not just a reviewer figure;

3) TTLL1 and TTLL4 mutants: it is important to include a discussion of why either one is required for development (arguing against redundancy), and also why some mutants make it to birth (some redundancy after all?).

4) The way the manuscript is written does not do a good service to the data included; it remains difficult to read and we hope the editorial staff can help with this.

Reviewer #3 (Remarks to the Author):

The Ms has been improved and most of the concerns previously mentioned have been addressed. The paper would still benefit from the inclusion of the data obtained in R1 ESCs (currently presented by the authors as "data not shown") and of the proliferation data currently shown in Figures 2C and 2D of the Response Letter.

Point-by-point response to the reviewers' comments

Reviewer #1 (Remarks to the Author):

The authors have addressed my concerns.

Reviewer #2 (Remarks to the Author):

The authors have address most of the concerns we raised. The points below should still be addressed:

1. Regarding the phenotype of the Klf4 point mutant:

a. 5G; labelling is misleading; "alive at E3.5" is not necessarily normal development - stage(s) that the embryos reached at 3.5 should be reported;

Answer: We changed it in the new Figure 5g.

b. "Primitive endoderm" is the correct designation, not "primary endoderm";

Answer: We changed it.

c. 5H, I: the quantification is convincing, but the images shown are of poor quality; can better images be show?

Answer: We provided better images in the new Fig. 5h, i.

2. On the important issue that the Klf4 point-mutant can pull down Klf5, suggestive of a potential reason for the more severe phenotype than Klf4^{-/-} mice: these data need to be discussed and show in a supplementary figure, not just a reviewer figure;

Answer: We showed the data in the new Supplementary Figure 5p and discussed the phenotype of Klf4 point-mutant mice accordingly.

3) TTLL1 and TTLL4 mutants: it is important to include a discussion of why either one is required for development (arguing against redundancy), and also why some mutants make it to birth (some redundancy after all?).

Answer: We discussed this issue in the discussion section.

4) The way the manuscript is written does not do a good service to the data included; it remains difficult to read and we hope the editorial staff can help with this.

Answer: We carefully revised our whole manuscript.

Reviewer #3 (Remarks to the Author):

The Ms has been improved and most of the concerns previously mentioned have been addressed. The paper would still benefit from the inclusion of the data obtained in R1 ESCs (currently presented by the authors as "data not shown") and of the proliferation data currently shown in Figures 2C and 2D of the Response Letter.

Answer: We provided R1 ESC data in the new Supplementary Figure 2c. We also provided these proliferation data in the new Supplementary Figure 1h and 2d.